# Timescales of chemical equilibrium between the convecting solid mantle and over-/underlying magma oceans

Daniela P. Bolrão[1], Maxim D. Ballmer[2,1], Adrien Morison[3], Antoine B. Rozel[1], Patrick Sanan[1], Stéphane Labrosse[3], and Paul J. Tackley[1]

[1]Institute of Geophysics, ETH Zurich, 8092 Zurich, Switzerland
[2]Department of Earth Sciences, University College London, London, WC1E 6BT, United Kingdom
[3]Université de Lyon, ENSL, UCBL, CNRS, LGL-TPE, 46 allée d'Italie, F-69364 Lyon, France

**Correspondence:** Daniela P. Bolrão (daniela.bolrao@erdw.ethz.ch)

**Abstract.** After accretion and formation, terrestrial planets go through at least one magma ocean episode. As the magma ocean crystallises, it creates the first layer of solid rocky mantle. Two different scenarios of magma ocean crystallisation involve that the solid mantle either (1) first appears at the core-mantle boundary and grows upwards, or (2) appears at mid-mantle depth and grows in both directions. Regardless of the magma ocean freezing scenario, the composition of the solid mantle and liquid reservoirs continuously change due to fractional crystallisation. This chemical fractionation has important implications for the long-term thermo-chemical evolution of the mantle, as well as its present-day dynamics and composition. In this work we use numerical models to study convection in a solid mantle bounded at either or both boundaries by magma ocean(s), and in particular, the related consequences for large-scale chemical fractionation. We use a parameterisation of fractional crystallisation of the magma ocean(s) and (re-)melting of solid material at the interface between these reservoirs. When these crystallisation/re-melting processes are taken into account, convection in the solid mantle occurs readily and is dominated by large wavelengths. Related material transfer across the mantle magma-ocean boundaries promotes chemical equilibrium, and prevents extreme enrichment of the last-stage magma ocean (as would otherwise occur due to pure fractional crystallisation). The timescale of equilibration depends on the convective vigour of mantle convection and on the efficiency of material transfer between the solid mantle and magma ocean(s). For Earth, this timescale is comparable to that of magma ocean crystallisation suggested in previous studies (Lebrun et al., 2013), which may explain why the Earth's mantle is rather homogeneous in composition, as supported by geophysical constraints.

*Copyright statement.* TEXT

## 1 Introduction

The early Earth experienced at least one episode of extensive silicate melting, also known as magma ocean (e.g., Abe and Matsui, 1988; Abe, 1993; Solomatov and Stevenson, 1993a; Abe, 1997; Solomatov, 2000; Drake, 2000; Elkins-Tanton, 2012). A magma ocean was likely formed due to the energy released during the Moon-forming giant impact (Tonks and Melosh, 1993;

Ćuk and Stewart, 2012; Canup, 2012), core formation (Flasar and Birch, 1973), radiogenic heating (Urey, 1956), electromagnetic induction heating (Sonett et al., 1968), and tidal heating (Sears, 1992). Due to the presence of an early atmosphere (Abe and Matsui, 1986; Hamano et al., 2013), it was sustained for thousands (Solomatov, 2000) to millions of years (Abe, 1997; Lebrun et al., 2013; Salvador et al., 2017; Nikolaou et al., 2019).

As the magma ocean cools and its temperature drops below the liquidus, crystals start to appear and consolidate a first layer of solid cumulates, i.e., the solid part of the mantle. Because the shape of the liquidus (and solidus) relative to the isentropic temperature profile is not well constrained, the depth at which initial crystallisation occurs remains unknown: this depth may be anywhere between the Core-Mantle Boundary (CMB) (e.g., Abe, 1997; Solomatov, 2015), and mid-mantle depths (Labrosse et al., 2007; Stixrude et al., 2009; Nomura et al., 2011; Labrosse et al., 2015; Boukaré et al., 2015; Caracas et al., 2019). Depending on this depth, several distinct scenarios of magma ocean evolution are expected to occur.

## 1.1 Crystallisation of a magma ocean from the bottom

If crystallisation of the magma ocean starts at the CMB, the first layer of solid cumulates forms at the bottom of this magma ocean (Fig. 1a). As the temperature of the ocean decreases, the crystallisation front steadily progresses upwards, creating more and more solid cumulates. When the crystallisation front reaches the surface of the planet, the solid mantle of the Earth is fully formed.

Assuming that the temperature of solid cumulates stays close to that of the solidus, these solid cumulates are thermally unstable since the solidus is steeper than the isentrope. Assuming as well that some degree of fractional crystallisation occurs (Solomatov and Stevenson, 1993b; Brown et al., 2014; Elkins-Tanton et al., 2003), the magma ocean becomes progressively enriched in iron silicates (FeO), since iron behaves like a mildly incompatible element (Murakami and Bass, 2011; Nomura et al., 2011; Andrault et al., 2012; Tateno et al., 2014). Accordingly, the solid cumulates (initially enriched in MgO) that form in chemical equilibrium with the overlying magma ocean incorporate progressively more FeO with time and, as a result, become denser with time (Fig. 1a). Therefore, on top of being thermally unstable, the solid cumulates are also gravitationally unstable due to composition. This leads to a large-scale overturn after magma ocean crystallisation (Elkins-Tanton et al., 2003, 2005), or multiple small-scale overturns during crystallisation (Maurice et al., 2017; Ballmer et al., 2017b; Boukaré et al., 2018; Morison et al., 2019; Miyazaki and Korenaga, 2019b). Such overturn(s) may lead to re-melting of FeO-enriched material at depth, as the isentrope of such material is steeper than its melting curve through most of the mantle.

This dense remelted material may form a Basal Magma Ocean (BMO) (Labrosse et al., 2015), join an already existing one (Labrosse et al., 2007) (see below), or react with the underlying solid mantle (Ballmer et al., 2017b). Hence, the solid mantle may evolve from being bounded above by only one magma ocean, the Top Magma Ocean (TMO), towards being bounded by two magma oceans, TMO and BMO, depending on the fate of overturned cumulates. Ultimately, the TMO is expected to completely crystallise, potentially leaving a long-lived BMO after the final overturn of the most FeO-enriched cumulates. Because the overturning events are potentially swift and of large scale nature, the resulting solid mantle and magma oceans are not necessarily in chemical equilibrium.

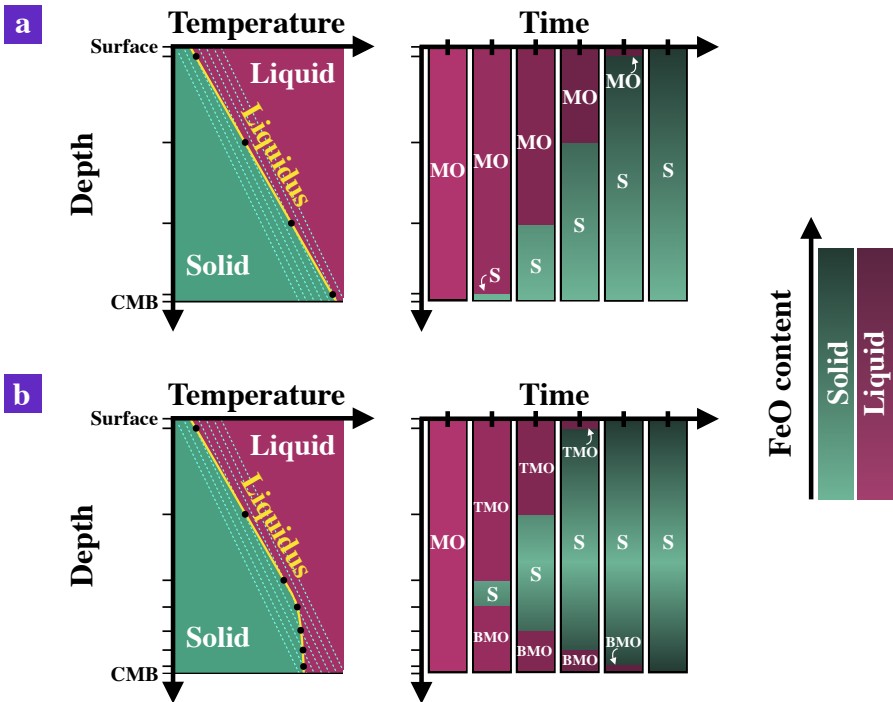

**Figure 1.** Sketches of Magma Ocean (MO) crystallisation scenarios. As cooling of the MO proceeds, adiabats (cyan dashed lines) cross the liquidus (yellow curve), and the Solid mantle (S) appears, either a) near the Core-Mantle Boundary (CMB), or b) somewhere at mid-mantle depths. In b the MO is divided in Top Magma Ocean (TMO) and Basal Magma Ocean (BMO) as soon as the solid appears. In both scenarios, liquid and solid cumulates get enriched in FeO with time, which may lead to an overturn of solid material (not depicted, see text for details). Mush is not considered.

## 1.2 Crystallisation of a magma ocean from the middle

If crystallisation of the magma ocean instead starts somewhere at mid-mantle depths and the crystals formed are near-neutrally buoyant (Labrosse et al., 2007; Boukaré et al., 2015), the first layer of solid mantle forms and separates the magma ocean into TMO and BMO (Fig. 1b). Then, two crystallisation fronts move in opposite directions: the TMO front progresses upwards until it reaches the surface of the planet, and the BMO front progresses downwards until it reaches the CMB. In this process, both TMO and BMO, as well as the related cumulates, become progressively enriched in FeO (Fig. 1b). In contrast to the TMO cumulates (see above), BMO cumulates are likely formed over much longer timescales (Labrosse et al., 2007) and are expected to form a stable density profile. By the time the BMO is fully crystallised, a dense stable solid layer may persist at the base of the mantle. This dense layer may explain seismic observations that point to the existence of thermo-chemical piles near the CMB (Masters et al., 2000; Ni and Helmberger, 2001; Garnero and McNamara, 2008; Deschamps et al., 2012; Labrosse et al., 2015; Ballmer et al., 2016).

### 1.3 Motivation

Along these lines, the chemical evolution of the solid mantle depends on the history of early planetary melting and crystallisation. This is a history with either one or two magma oceans, and with convection in the solid mantle driven by unstable thermal and/or chemical stratification, probably while magma ocean(s) at the top and/or bottom are still present. While any such convection would imply re-melting of solid cumulates, the related consequences for mantle evolution are poorly understood. Only a few numerical modelling studies have explicitly incorporated coupled re-melting and crystallisation at the magma ocean mantle boundary or boundaries (Labrosse et al., 2018; Morison et al., 2019; Agrusta et al., 2019), and none of these studies have explored the consequences for chemical evolution.

In this paper we use a numerical model to investigate the thermo-chemical evolution of the solid mantle in contact with a TMO and/or a BMO. We consider that convection in the solid mantle starts before the end of magma ocean crystallisation, therefore, dynamic topographies that may form at either or both solid mantle-magma ocean boundaries can melt or crystallise. We do not explicitly account for the progression of the crystallisation front(s). However, we test several evolution scenarios and different magma oceans thicknesses. We determine the timescales of chemical equilibrium between the magma ocean(s) and the solid mantle, and compare them with those of progression of the crystallisation front (e.g., Lebrun et al., 2013). For simplicity, we hereafter use the term *solid-liquid phase changes* interchangeably with fractional crystallisation and melting processes at the interface between the solid mantle and TMO and/or BMO.

## 2 Numerical model

### 2.1 Problem definition

We use the finite-volume/finite-difference method with the convection code StagYY (Tackley, 2008), to model the thermo-chemical evolution of the solid mantle during magma ocean crystallisation. We test three different evolution scenarios, as the solid mantle may be bounded above by a TMO and/or below by a BMO (Fig. 2). We assume steady crystallisation front(s) and test different magma ocean thicknesses: when only one ocean is present, it can be 100, 500 or 1000 km thick; when both oceans are present, the thickness of each ocean is 100 and/or 500 km.

We assume that the solid mantle is an infinite Prandtl number fluid. We assume mechanical stability between the solid mantle and magma oceans, i.e., $\rho_{\text{TMO}} < \rho_{\text{S}} < \rho_{\text{BMO}}$, with $\rho_{\text{TMO}}$, $\rho_{\text{S}}$ and $\rho_{\text{BMO}}$ the densities of the TMO, solid mantle and BMO, respectively. We take gravitational acceleration, $g$, viscosity, $\eta$, thermal diffusivity, $\kappa$, heat capacity, $C_p$, thermal expansion coefficient, $\alpha$, and compositional expansion coefficient, $\beta$, as constant. Values of these parameters can be found in Table 1.

We make equations dimensionless to reduce the number of parameters that describe the physical problem. Dimensions of distance, time and temperature can be recovered using, respectively, the thickness of the solid mantle, $h_{\text{S}}$, the thermal diffusive timescale, $\frac{h_{\text{S}}^2}{\kappa}$, and the temperature difference between bottom and top solid mantle boundaries, $\Delta T = T^- - T^+$. The

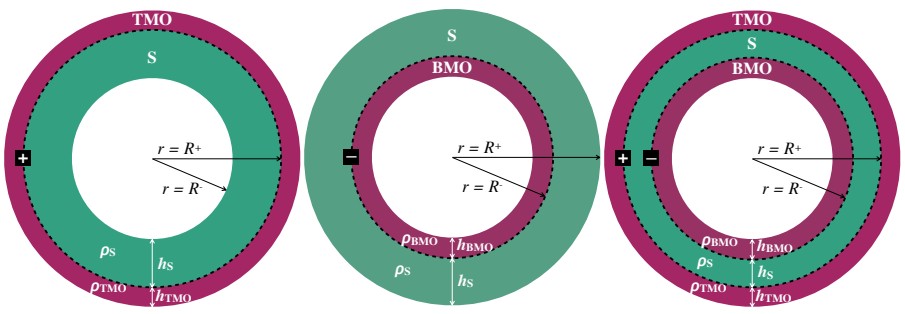

**Figure 2.** Sketches of three possible evolution scenarios: Solid mantle (S) bound by left) Top Magma Ocean (TMO), middle) Basal Magma Ocean (BMO) and right) TMO and BMO. Solid mantle is taken as a spherical shell with density $\rho_S$, thickness $h_S$, inner radius $R^-$, and outer radius $R^+$. TMO and BMO are taken with densities $\rho_{TMO}$ and $\rho_{BMO}$, and thicknesses $h_{TMO}$ and $h_{BMO}$, respectively. Superscripts "$+$" and "$-$" at the boundaries refer to the boundary between TMO and solid mantle, and solid mantle and BMO, respectively.

**Table 1.** Parameters used in the simulations.

| Parameter [dimension] | Symbol | Value |
|---|---|---|
| Radius of the Core-Mantle Boundary [km] | $R_{CMB}$ | 3480 |
| Total radius of the planet [km] | $R_P$ | 6370 |
| Thickness of the solid mantle today [km] | $h_M$ | 2890 |
| Gravitational acceleration [m s$^{-2}$] | $g$ | 9.81 |
| Viscosity [Pa s] | $\eta$ | - |
| Thermal diffusivity [m$^2$s$^{-1}$] | $\kappa$ | $5 \times 10^{-7}$ |
| Heat capacity [J kg$^{-1}$ K$^{-1}$] | $C_p$ | 1200 |
| Thermal conductivity [W m$^{-1}$ K$^{-1}$] | $k$ | 3.0 |
| Thermal expansion coefficient [1/K] | $\alpha$ | $10^{-5}$ |
| Reference density [kg m$^{-1}$] | $\rho$ | 5000 |
| Buoyancy number [-] | B | 1.0 |
| Solid/liquid partition coefficient of FeO [-] | $K$ | 0.3 |
| Phase change number [-] | $\Phi$ | $10^{-1} - 10^5$ |
| Thickness of Top Magma Ocean [km] | $h_{TMO}$ | 0 if only BMO present, 100, 500, 1000 |
| Thickness of Basal Magma Ocean [km] | $h_{BMO}$ | 0 if only TMO present, 100, 500, 1000 |
| Super-criticality [-] | SC | $10^2 - 10^5$ |
| FeO concentration of the bulk [-] | $X_{FeO}^{Bulk}$ | 0.12 |

dimensionless temperature, $T$, is defined as:

$$T = \frac{T' - T^+}{\Delta T} \tag{1}$$

We assume incompressibility in the Boussinesq approximation (e.g., Chandrasekhar, 1961). Therefore mass, energy, composition and momentum conservation equations are written as:

$$\boldsymbol{\nabla} \cdot \mathbf{u} = 0 \tag{2}$$

$$\frac{\partial T}{\partial t} + \mathbf{u} \cdot \boldsymbol{\nabla} T = \nabla^2 T \tag{3}$$

$$\frac{\partial X_{\text{FeO}}^{\text{S}}}{\partial t} + \mathbf{u} \cdot \boldsymbol{\nabla} X_{\text{FeO}}^{\text{S}} = 0 \tag{4}$$

$$-\boldsymbol{\nabla} p + \nabla^2 \mathbf{u} + \text{Ra}\Big(T - \langle T \rangle - \text{B}(X_{\text{FeO}}^{\text{S}} - \langle X_{\text{FeO}}^{\text{S}} \rangle)\Big)\hat{\mathbf{r}} = 0 \tag{5}$$

with $\mathbf{u}$ the velocity field, $\langle T \rangle$ the lateral average of the temperature field $T$, $t$ the time, $X_{\text{FeO}}^{\text{S}}$ the FeO molar content in the solid mantle, $\langle X_{\text{FeO}}^{\text{S}} \rangle$ the lateral average of $X_{\text{FeO}}^{\text{S}}$, $p$ the dynamic pressure, Ra the Rayleigh number and B the buoyancy number. The last two are defined respectively as:

$$\text{Ra} = \frac{\rho g \alpha \Delta T h_{\text{S}}^3}{\eta \kappa}, \tag{6}$$

$$\text{B} = \frac{\beta}{\alpha \Delta T}. \tag{7}$$

In this study we consider that magma oceans and solid mantle are made only of (Fe, Mg)O (see section 2.3 for more details). We set temperature to 1.0 and 0.0, respectively, at the bottom and top solid domain boundaries. Regarding the buoyancy number, Earth-like models point to a value of $\text{B} \approx 3$ for the present-day mantle (i.e., based on the density difference between Mg- and Fe-rich silicate end-members, as well as CMB temperature estimates). However, there are significant uncertainties associated with the value of $\alpha$ (which is temperature and pressure dependent, e.g., Tosi et al., 2013), and that of $\Delta T$ in the solid part of the primitive mantle. For example, $\Delta T$ increases with the thickness of the solid layer. Within these uncertainties, at least a range of $1 \leq \text{B} \leq 3$ is acceptable. In this paper, we choose $\text{B} = 1.0$ in order to limit the impact of the initial condition on the onset of convection (see discussion in Section 4).

The solid domain is represented using the spherical annulus geometry (Hernlund and Tackley, 2008), composed of a grid of $512 \times 128$ cells, in which Eq. (2) – Eq. (5) are solved. Composition is advected by tracers. We assume that each magma ocean is well-mixed and that its dynamics are fast compared to that of the solid mantle. In our setup, magma oceans are treated as simple 0D compositional reservoirs at solid mantle boundaries. We hereafter use superscripts '+' and '−' to refer, respectively, to top and bottom solid mantle boundaries. In equations, the sign '±' reads as '+' if a TMO is considered, and '−' if a BMO is considered. The subscript 'MO' refers to Magma Ocean. Thus, when we introduce a quantity, e.g. $\xi$, related to a magma ocean, we introduce it as $\xi_{\text{MO}}^{\pm}$, with $\xi_{\text{MO}}^{+} = \xi_{\text{TMO}}$ relating to the TMO, and $\xi_{\text{MO}}^{-} = \xi_{\text{BMO}}$ relating to the BMO.

## 2.2 Dynamic topography and the phase change boundary condition

Since convection in the solid mantle likely starts before the end of magma ocean crystallisation (Maurice et al., 2017; Ballmer et al., 2017b; Boukaré et al., 2018; Morison et al., 2019; Miyazaki and Korenaga, 2019b), dynamic topographies are supported at either or both solid mantle boundaries. The timescale for producing dynamic topography is noted $\tau_\eta$. This topography can be eroded by solid-liquid phase changes on a timescale related to the transfer of energy and FeO through the magma ocean, from material that is crystallising to material that is melting. We denoted this timescale by $\tau_\phi$.

The relative values of the two timescales, $\tau_\eta$ and $\tau_\phi$, control the dynamical behaviour of the boundary. If $\tau_\eta \ll \tau_\phi$, dynamic topography can build before being erased by the phase change. In this case, dynamic topography is only limited by the balance between viscous stress in the solid and the buoyancy associated with the topography. In the limit of small topographies, this leads to the classical non-penetrating free-slip boundary condition in which the radial velocity of the solid effectively goes to 0 at the boundary (Ricard et al., 2014). On the other hand, if $\tau_\eta \gg \tau_\phi$, the topography is erased faster by phase changes than it can be built by viscous stress in the solid. Consequently, this removes the stress imposed by the topography and the associated limit to the radial velocity. These processes are incorporated into our boundary condition, described by the phase change number,

$$\Phi = \frac{\tau_\phi}{\tau_\eta}, \tag{8}$$

considering that when $\Phi \to \infty$, dynamic topography is built (or relaxes) by viscous forces, and when $\Phi \to 0$ it is eroded by melting or fractional crystallisation processes (Deguen, 2013; Deguen et al., 2013). The related phase-change boundary condition in dimensionless form, at either or both solid mantle boundaries is:

$$2\frac{\partial u_r}{\partial r} - p \pm \Phi^\pm u_r = 0 \tag{9}$$

with $u_r$ the vertical velocity of the flow in the solid mantle. On one hand, this boundary condition can act like a non–penetrating free–slip boundary condition when $\Phi \to \infty$, since vertical velocities of the solid flow tend to 0 at the boundaries. Under this boundary condition, transfer of material across a solid mantle-magma ocean boundary cannot occur. On the other hand, this boundary condition can act as being "open" to phase changes when $\Phi \to 0$, since these vertical velocities will be non zero at the solid mantle-magma ocean boundary, and a significant flux of solid and liquid material can cross it to melt and crystallise. Hence, transfer of material across the phase-change boundary is efficient. In the extreme case of $\Phi = 0$, this boundary condition corresponds to free in- and outflow.

The specific value of $\Phi$ is difficult to constrain (because $\tau_\phi$ is non-trivial to determine), and also is expected to vary with time (i.e., because $\tau_\eta$ depends on the thickness of the solid mantle) (Deguen, 2013; Deguen et al., 2013). However, for a purely thermal case, Morison et al. (2019) and Morison (2019) estimate $\Phi^+ \sim 10^{-5}$ and $\Phi^- \sim 10^{-3}$ for the Earth. Therefore, significant transfer of material across the solid-mantle magma-ocean boundaries is expected. However, also consider that real multi-phase rocks typically melt over large pressure ranges, unless for truly eutectic bulk compositions. The depleted residue of mantle melting may restrict the efficiency of material transfer across the solid mantle magma-ocean boundaries, depending on the efficiency of melt-solid segregation near the boundaries. In addition to the expected temporal evolution of $\Phi^\pm$, this potential restriction motivates our exploration of a broad range of $\Phi^\pm$. In this study we use 7 values of $\Phi^\pm$ that range from

$10^{-1}$ to $10^{5}$. We use $\Phi = 10^{-1}$ as the lowest value possible for $\Phi^{\pm}$ because the resolution of the thermal boundary layer is computationally demanding once $\Phi^{\pm}$ decreases below $10^{-1}$.

Deguen et al. (2013) and Labrosse et al. (2018) found that the critical Rayleigh number, $Ra_c$, for the solid mantle is strongly
sensitive to $\Phi$ and the setup considered, i.e., having a TMO and/or a BMO, as well as to the thickness of the solid layer. For instance, if the solid mantle is bounded by a TMO of 100 km and $\Phi \to \infty$, $Ra_c$ is on the order of $10^{3}$, but for small $\Phi$, is on the order of $10^{2}$. $Ra_c$ can even decrease to arbitrarily small values on the order of $\sim \Phi$ if a TMO and BMO are both considered. Therefore, we also systematically vary the Rayleigh number, Ra, which controls the convective vigour of the mantle. We choose Ra as multiples of $Ra_c$, according to the super-criticality factor, SC:

$\mathrm{Ra} = \mathrm{Ra}_c \times \mathrm{SC}.$                                                        (10)

We use 4 values of SC ranging from $10^{2}$ to $10^{5}$.

## 2.3   Compositional treatment

In this study we consider a simplified compositional model with only two components, FeO and MgO, which are thought to be the Fe-rich and Mg-rich end-members of mantle silicates. We denote the FeO and MgO molar content in the solid and magma
ocean parts, respectively, by $X_{\mathrm{FeO}}^{\mathrm{S}}$ and $X_{\mathrm{FeO}}^{\mathrm{MO}}$, and $X_{\mathrm{MgO}}^{\mathrm{S}}$ and $X_{\mathrm{MgO}}^{\mathrm{MO}}$. We consider mass balance between FeO and MgO in the solid mantle and magma oceans, therefore, $X_{\mathrm{FeO}}^{\mathrm{S}} + X_{\mathrm{MgO}}^{\mathrm{S}} = 1$ and $X_{\mathrm{FeO}}^{\mathrm{MO}} + X_{\mathrm{MgO}}^{\mathrm{MO}} = 1$.

Our model simulates melting and crystallisation of material depending on the influx and outflux of material at the solid mantle boundary. Melting of solid material is simulated when dynamic topography develops outside the solid domain, i.e., when there is an outflux of material of the solid domain. It is assumed that no fractionation operates when the solid melts, i.e.,
all (Fe,Mg)O present in this topography goes into the magma ocean. Therefore, tracers that leave the solid domain pass their information (about mass and composition) to the magma ocean, and are deleted.

We simulate crystallisation of the magma ocean when negative dynamic topography develops in the solid domain, i.e., when there is an influx of mass in the solid domain. When this happens, the influx of material pushes tracers and cells near the boundary are left with no tracers. To ensure mass conservation, new $n$ tracers are introduced in those cells, which simulates
solid mantle being created. To determine $n$, i.e., the amount of tracers that need to be introduced in the solid part, we calculate the influx of mass corresponding to this dynamic topography, and divide it by the tracer ideal mass. We then equally distribute the influx of mass by $n$ tracers. The composition of the solid created is related to that of the liquid by fractional crystallisation, therefore, only a fraction of FeO goes into the solid. This fraction is given by the distribution coefficient, $K$:

$$K = \frac{X_{\mathrm{FeO}}^{\mathrm{S}} X_{\mathrm{MgO}}^{\mathrm{MO}}}{X_{\mathrm{FeO}}^{\mathrm{MO}} X_{\mathrm{MgO}}^{\mathrm{S}}}.$$                                       (11)

We assume $K = 0.3$ (e.g., Corgne and Wood, 2005; Liebske et al., 2005). The difference between influx and outflux of material through the boundary is of the order of $10^{-15}$, meaning that conservation of mass in the solid mantle is ensured.

In this paper, we attempt to estimate the characteristic timescale to establish chemical equilibrium between the solid mantle and the magma ocean(s). Assuming a full equilibrium between the solid mantle and magma oceans (superscript "Eq"), the FeO

content in the bulk, $X_{\text{FeO}}^{\text{Bulk}}$, can be expressed as function of the volumes ($V_{\text{S}}$, $V_{\text{TMO}}$ and $V_{\text{BMO}}$) and the FeO content ($X_{\text{FeO}}^{\text{S,Eq}}$, $X_{\text{FeO}}^{\text{TMO,Eq}}$ and $X_{\text{FeO}}^{\text{BMO,Eq}}$) in the solid mantle and magma oceans,

$$X_{\text{FeO}}^{\text{Bulk}} = \frac{X_{\text{FeO}}^{\text{TMO,Eq}}V_{\text{TMO}} + X_{\text{FeO}}^{\text{BMO,Eq}}V_{\text{BMO}} + X_{\text{FeO}}^{\text{S,Eq}}V_{\text{S}}}{V_{\text{TMO}} + V_{\text{BMO}} + V_{\text{S}}}. \tag{12}$$

From Eq. (11) and Eq. (12) one can find the FeO content in the solid mantle when it is in chemical equilibrium with the magma ocean(s):

$$X_{\text{FeO}}^{\text{S, Eq}} = \frac{-b + \sqrt{b^2 - 4ac}}{2a}, \tag{13}$$

where:

$$a = V_{\text{S}}(1 - K),$$
$$b = V_{\text{TMO}} + V_{\text{BMO}} + V_{\text{S}}K - X_{\text{FeO}}^{\text{Bulk}}(V_{\text{TMO}} + V_{\text{BMO}} + V_{\text{S}})(1 - K),$$
$$c = -X_{\text{FeO}}^{\text{Bulk}}K(V_{\text{TMO}} + V_{\text{BMO}} + V_{\text{S}}).$$

But, because chemical equilibrium would take too long to reach in a reasonable run time, we look for the timescale to reach chemical half-equilibrium. Starting with a FeO content in the solid mantle equal to $X_{\text{FeO}}^{\text{S, Ini}}$, the half-equilibrium is reached when the solid mantle reaches the content $X_{\text{FeO}}^{\text{S, Eq/2}}$, defined as:

$$X_{\text{FeO}}^{\text{S, Eq/2}} = \frac{X_{\text{FeO}}^{\text{S, Ini}} + X_{\text{FeO}}^{\text{S,Eq}}}{2}. \tag{14}$$

We denote by $t^{\text{S, Eq/2}}$ the time at which the solid mantle reaches chemical half-equilibrium, $t^{\text{S, Eq/2}} = t(X_{\text{FeO}}^{\text{S, Eq/2}})$.

Previous studies suggest that the Fe content of the present day bulk silicate Earth is 0.113 (Taylor and McLennan, 1985) or 0.107 (McDonough and Sun, 1995). We suppose that some of the Fe could migrate to the core with time (e.g., Nguyen et al., 2018) and therefore, in this study we use $X_{\text{FeO}}^{\text{Bulk}} = 0.120$. We start the simulations with a homogeneous FeO content in the solid mantle and magma ocean(s), $X_{\text{FeO}}^{\text{S, Ini}} = X_{\text{FeO}}^{\text{TMO, Ini}} = X_{\text{FeO}}^{\text{BMO, Ini}} = 0.120$. Although this initial composition is not consistent with the fractional crystallisation assumed in this problem, it serves well our goal of measuring the timescale to reach chemical equilibrium between solid and liquid reservoirs.

# 3 Results

## 3.1 Chemical evolution of the mantle bounded on top by a TMO

In this subsection we investigate how the chemical evolution of the solid mantle is affected by the efficiency of mass transfer across the phase-change boundary, as controlled by $\Phi$. As mentioned in the previous section, low values of $\Phi$ correspond to efficient material transfer across the phase-change boundaries, and high values of $\Phi$ correspond to inefficient material transfer, similar to classical convection. We analyse first the case of a solid mantle bound above by a TMO, as the most straightforward

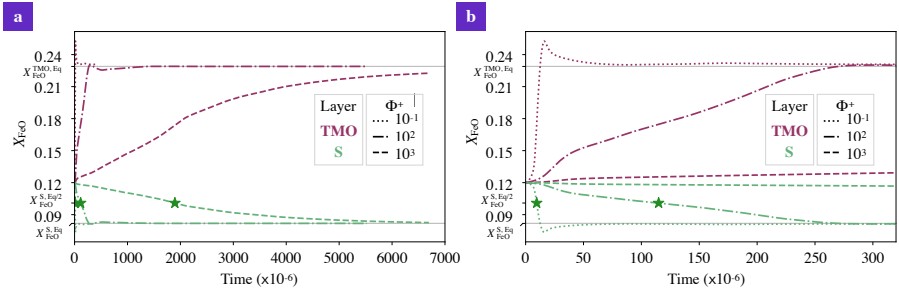

**Figure 3.** a) Evolution of the FeO content, $X_{\text{FeO}}$, in the solid mantle (S, green) and top magma ocean of 500 km (TMO, pink), with dimensionless time units. We test three different values of phase change number, $\Phi^+ = 10^{-1}, 10^2, 10^3$ (different lines). Green stars correspond to the point where FeO content in chemical half-equilibrium in the solid mantle, $X_{\text{FeO}}^{\text{S, Eq/2}}$, is reached. In these simulations super-criticality is SC $= 10^5$. b) Zoom of figure a) in the beginning of evolution.

scenario for early planetary evolution. In the end of this subsection we briefly compare this scenario with the ones where the solid mantle is in contact with just a BMO and with both magma oceans.

Because the parameter space explored in this paper is vast, we illustrate here as an example the chemical evolution of a solid mantle bounded by a TMO of 500 km, under three different values of phase change number, $\Phi^+ = 10^{-1}, 10^2, 10^3$, at the same

super-criticality value of SC $= 10^5$. As the critical Rayleigh number, $\text{Ra}_c$, decreases as $\Phi^+$ decreases, these three cases have different values of Rayleigh number, Ra. Hence, for $\Phi^+ = 10^{-1}, 10^2, 10^3$, $\text{Ra} = 100 \times 10^5, 635 \times 10^5, 687 \times 10^5$, respectively. According to Eq. (12) and Eq. (13), the expected FeO content in each reservoir in chemical equilibrium would be approximately $X_{\text{FeO}}^{\text{S,Eq}} = 0.082$ and $= X_{\text{FeO}}^{\text{TMO,Eq}} = 0.229$. Since we initialise each reservoir with a FeO content of $X_{\text{FeO}}^{\text{S, Ini}} = X_{\text{FeO}}^{\text{TMO, Ini}} = 0.120$, the system does not start in chemical equilibrium. We determine the time needed to reach chemical half-equilibrium.

Figure 3 shows the chemical evolution in dimensionless time units of these three cases. Our models predict that regardless of the value of $\Phi^+$, the FeO content in the solid mantle decreases towards $X_{\text{FeO}}^{\text{S,Eq}}$, and the FeO content in the TMO increases towards $X_{\text{FeO}}^{\text{TMO,Eq}}$, thereby bringing the solid mantle and the TMO close to chemical equilibrium (but not chemical homogeneity as seen later). However, the lower the value of $\Phi^+$, the faster half-equilibrium is reached, since it effectively increases the exchange of material between reservoirs. We calculate the time needed to reach chemical half-equilibrium, and for $\Phi = 10^{-1}$,

half-equilibrium is reached $\sim 10$ times faster than for $\Phi = 10^2$, and $\sim 200$ times faster than for $\Phi = 10^3$.

In Fig. 4 we present snapshots of FeO content in the solid mantle for these three cases. Our models show that dynamics in the solid mantle is very different between cases. With $\Phi^+ = 10^{-1}$ (Fig. 4a), mantle flow is dominated by degree-1 convection, which persists stably for the whole simulation time. With this pattern of convection, there is an upwelling of primordial material (in yellow) that melts on one hemisphere, while material from the TMO crystallises at the boundary and forms a

235 downwelling on the other hemisphere (in blue). This downwelling is FeO depleted, which introduces a strong heterogeneity in the solid mantle. Degree-1 convection involves very little deformation, which explains the existence of a considerable amount of primordial material in the solid mantle, even around the time at which chemical half-equilibrium occurs (snapshot inside red box). As $\Phi^+$ increases (Fig. 4b and Fig. 4c), higher degree modes of convection with several convection cells appear. Although

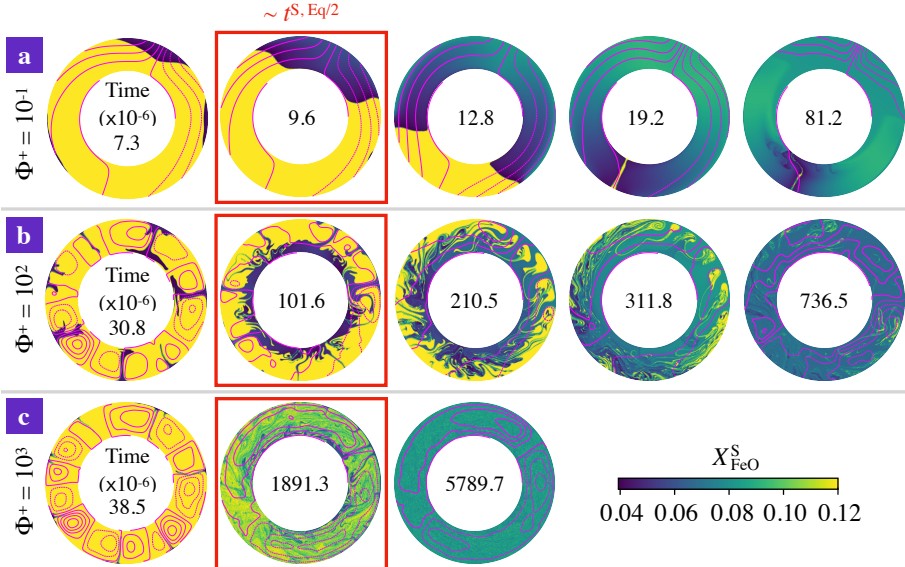

**Figure 4.** Snapshots of FeO content in the solid mantle, $X_{FeO}^S$, as function of dimensionless time (factor of $10^{-6}$ inside each annulus) for the cases presented in Fig. 3. a) $\Phi^+ = 10^{-1}$, b) $\Phi^+ = 10^2$ and c) $\Phi^+ = 10^3$. In these simulations the top magma ocean thickness is 500 km, and there is no basal magma ocean. Super-criticality is $SC = 10^5$ in all three cases shown. Snapshots with a red box indicate that the model time is close to the chemical half-equilibrium time, $t^{S, Eq/2}$. For $\Phi^+ = 10^{-1}$, $10^2$ and $10^3$, $t^{S, Eq/2} = 9.6 \times 10^{-6}$, $114.9 \times 10^{-6}$ and $1895.5 \times 10^{-6}$, respectively. Magenta contours correspond to the streamlines of the flow.

the composition of the TMO and the average composition of the solid mantle tend to mutual chemical equilibrium in all three
cases, chemical homogeneity across the solid mantle is not necessarily reached.

Our models show that for other evolution scenarios, i.e., solid mantle in contact with just a BMO and with a TMO and BMO, the system also evolves to a state close to chemical equilibrium but not chemical homogeneity. In Fig. 5 we present snapshots of FeO content in the solid mantle for different evolution scenarios at about the time of chemical half-equilibrium. When the solid mantle is in contact with just a BMO, material from the magma ocean crystallises at the boundary and forms upwellings
(in blue). This material is FeO depleted and, similarly to the TMO case, introduces a strong heterogeneity in the solid mantle around the half-equilibrium time. When the solid mantle is in contact with both oceans, convection occurs with degree-1, i.e., material of the TMO and of the BMO crystallises at the corresponding boundary and forms a downwelling (in blue) and an upwelling (in blue and green), respectively. Note that in this scenario, since the volume of the BMO is smaller than that of the TMO, the BMO composition changes rapidly. Therefore, the composition of the upwelling changes rapidly as well (colours
from blue to green). The degree-1 pattern of convection persists stably for the whole simulation time.

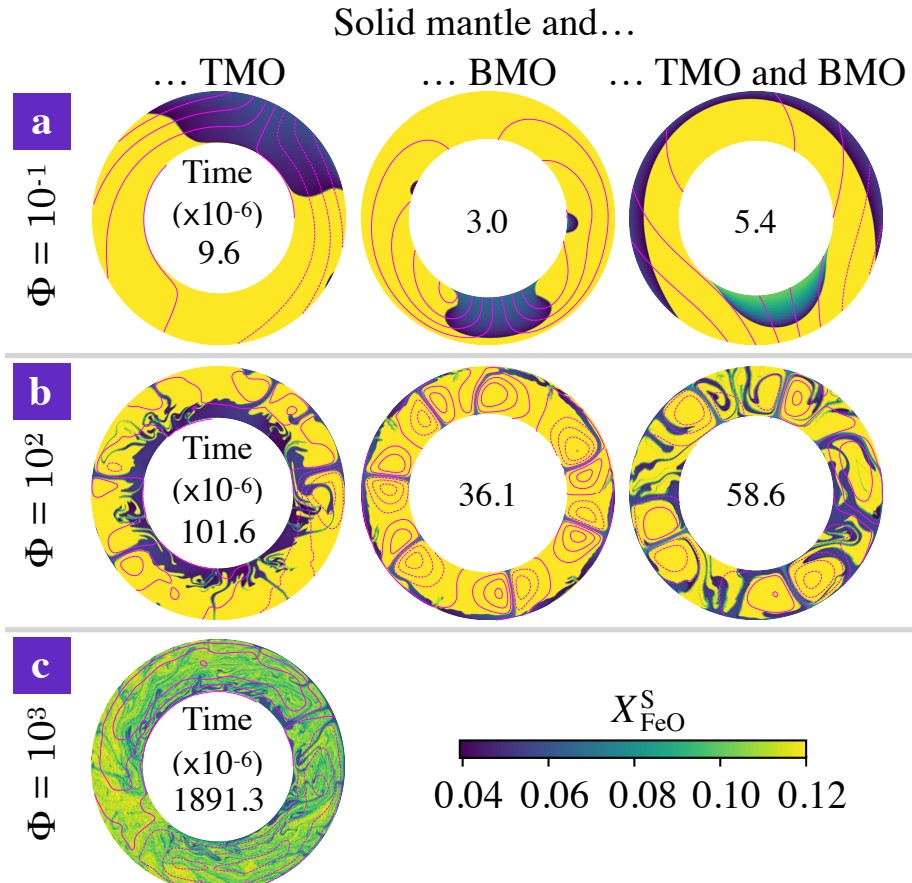

**Figure 5.** Snapshots of FeO content in the solid mantle, $X_{FeO}^S$, close to the time at which the system reaches chemical half-equilibrium (time is dimensionless with a factor of $10^{-6}$ inside each annulus), for a) $\Phi = 10^{-1}$, b) $\Phi = 10^2$ and c) $\Phi = 10^3$. The solid mantle is in contact with left) a top magma ocean (TMO) of 500 km, middle) a basal magma ocean (BMO) of 500 km, and right) a TMO of 500 km and a BMO of 100 km. Super-criticality is $SC = 10^5$ in all cases shown. Magenta contours correspond to the streamlines of the flow. Cases with $\Phi = 10^3$ for a solid mantle in contact with only BMO and with TMO and BMO, did not reach chemical half-equilibrium. Note that the difference between the aspect ratio of each evolution scenario is too small to be noticed in these annuli.

## 3.2 Timescales of chemical half-equilibrium between the solid mantle and magma ocean(s)

Figure 6 shows the timescales of chemical half-equilibrium for the scenarios explored in the previous subsection. For a wide range of SC and $\Phi^{\pm}$, these timescales are shown for a solid mantle bounded by a TMO of 500 km thickness (Fig. 6a), by a BMO of 500 km thickness (Fig. 6b), and by a TMO and a BMO of 500 km and 100 km thickness, respectively (Fig. 6c). For all evolution scenarios, models predict that timescales of chemical half-equilibrium decrease for decreasing $\Phi^{\pm}$. In other words, chemical half-equilibration is more efficient for efficient material transfer across the solid mantle-magma ocean boundaries.

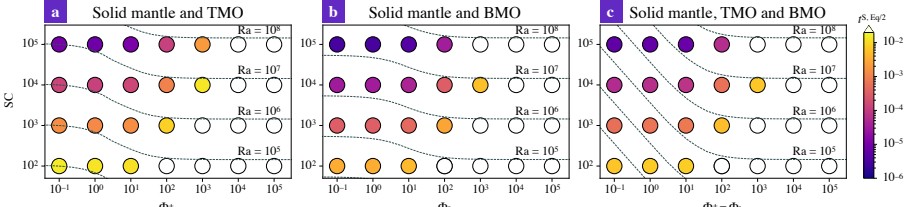

**Figure 6.** Timescales to reach chemical half-equilibrium, $t^{\mathrm{S,\,Eq/2}}$ (in different colours) between the solid mantle and a) a top magma ocean (TMO) of 500 km, b) a basal magma ocean (BMO) of 500 km, and c) a TMO of 500 km and BMO of 100 km, for different values of super-criticality, $\mathrm{SC} = 10^2 - 10^5$ (Ra is indicated by dashed lines), and phase change number, $\Phi = 10^{-1} - 10^5$. White circles are cases that did not reach chemical half-equilibrium within a reasonable run time.

Our results also show that the timescales of chemical half-equilibration are similar (i.e., of the same order of magnitude) for a given SC and $\Phi^{\pm}$ ranging between $10^{-1}$ and $10^1$, independent of the evolution scenario. This shows that below $\Phi^{\pm} = 10^1$ a regime with efficient material transfer across the solid mantle-magma ocean boundaries is established. The transition to the regime of inefficient material transfer (i.e., in which mantle flow is limited by viscous building of dynamic topography) occurs somewhere between $\Phi^{\pm} = 10^1$ and $10^2$. In this regime, timescales of half-equilibration systematically increase with $\Phi^{\pm}$. Our models predict that this transition between regimes occurs over a similar interval of $\Phi^{\pm}$ for other thicknesses of TMO and/or BMO.

To obtain an empirical scaling law, we fit the predicted timescales, $t_{\mathrm{pred}}^{\mathrm{S,\,Eq/2}}$, for all simulations that reached chemical half-equilibrium. The fitting equation provides $t_{\mathrm{pred}}^{\mathrm{S,\,Eq/2}}$ in dimensionless form as a function of Ra, $\Phi^{\pm}$ and $V_{\mathrm{S}}$:

$$t_{\mathrm{pred}}^{\mathrm{S,\,Eq/2}} = \max\left[a_0 \mathrm{Ra}^{a_1}\left(\frac{\Phi^{\pm}}{10}\right)^{a_2}\left(\frac{V_{\mathrm{S}}}{V_{\mathrm{M}}}\right)^{a_3}, a_4 \mathrm{Ra}^{a_5}\left(\frac{\Phi^{\pm}}{10}\right)^{a_6}\left(\frac{V_{\mathrm{S}}}{V_{\mathrm{M}}}\right)^{a_7}\right] \tag{15}$$

with $V_{\mathrm{M}}$ the volume of the present-day Earth's mantle. Coefficients of this equation can be found in Table 2. In the Appendix A of this paper, we explain the regression method and show a good agreement between the timescales to reach chemical half-equilibrium, obtained with our model predictions and our empirical scaling law (Fig. A1).

Equation (15) presents two branches, each corresponding to a different regime: the left branch corresponds to the regime of efficient material transfer across the solid mantle-magma ocean boundaries, and the right one to the regime of inefficient material transfer. Our models predict that in the regime of efficient material transfer (i.e., for low values of $\Phi$), timescales to reach chemical half-equilibrium are only loosely affected by the volume of the solid mantle (or in other words, by the volume of the magma ocean(s)). The volume of the solid only systematically affects the timescales once the regime shifts to the one of inefficient material transfer. This conclusion is independent of the evolution scenario. One possible explanation for the weak influence of the solid mantle's volume is that at low values of $\Phi$, convection occurs with low degree, so the geometry of the problem is less important.

We find that timescales to reach chemical half-equilibrium are about a factor of 3 larger for a solid mantle in contact with just a TMO (Fig. 6a) than for a solid mantle with just a BMO (Fig. 6b). This can be explained by the fact that the geometry of

**Table 2.** Results of the regressions of the timescales of chemical half-equilibration using the form: $t_{\text{pred}}^{\text{S, Eq/2}} = \max\left[a_0\text{Ra}^{a_1}\left(\frac{\Phi^\pm}{10}\right)^{a_2}\left(\frac{V_S}{V_M}\right)^{a_3}, a_4\text{Ra}^{a_5}\left(\frac{\Phi^\pm}{10}\right)^{a_6}\left(\frac{V_S}{V_M}\right)^{a_7}\right]$. The regression method is detailed in Appendix A.

| Regime | Coefficient | Solid mantle in contact with | | |
|---|---|---|---|---|
| | | TMO | BMO | TMO and BMO |
| Solid-liquid phase changes ($\Phi \to 0$) | $a_0$ | 464.850 | 103.146 | 100.473 |
| | $a_1$ | $-1.042$ | $-1.008$ | $-1.000$ |
| | $a_2$ | 0.313 | 0.176 | 0.948 |
| | $a_3$ | $-0.994$ | $-1.326$ | $-0.646$ |
| Viscous building ($\Phi \to \infty$) | $a_4$ | 12.075 | 20.743 | 48.481 |
| | $a_5$ | $-0.884$ | $-0.972$ | $-0.999$ |
| | $a_6$ | 1.214 | 1.208 | 1.195 |
| | $a_7$ | $-2.584$ | $-7.278$ | $-2.583$ |
| | error (%) | 28.3 | 21.2 | 22.4 |

the problem is different in both cases. Although the TMO and the BMO have the same thickness, the volume of the TMO is larger than that of the BMO by roughly a factor of 3, which explains the increased duration to reach the half-equilibrium FeO content in the magma ocean.

When it comes to a solid mantle bounded by both oceans (Fig. 6c), models predict that timescales are roughly 2 orders of magnitude smaller than the ones of a solid mantle in contact with just a TMO (Fig. 6a), for a given Rayleigh number. This result is explained by two effects. The critical Rayleigh number is much lower when two magma oceans are present than when only one is present. In principle, when both magma oceans are present, the critical Rayleigh number can even be arbitrarily low as $\Phi^\pm$ decreases towards 0 (Labrosse et al., 2018). Moreover, Agrusta et al. (2019) showed that the heat flow and RMS velocity in the solid mantle vary linearly with Ra when both magma oceans are present, whereas heat flow and RMS velocity in the solid mantle vary as $\text{Ra}^{1/3}$ and $\text{Ra}^{2/3}$, respectively, in the case of only one magma ocean present. This further increases the difference between the two scenarios at a given value of the Rayleigh number. Therefore, one should expect that the timescales to reach chemical half-equilibrium may be arbitrarily low, depending on the efficiency of material transfer across the BMO-mantle and TMO-mantle boundaries.

### 3.3 Chemical half-equilibrium and crystallisation timescales

In this subsection we compare the timescales to reach chemical half-equilibrium between a solid mantle and a TMO of a given thickness, with timescales of crystallisation of such a TMO as calculated for the Earth case. The timescales of TMO crystallisation (i.e., before reaching the mush stage) are given by Lebrun et al. (2013), hereinafter denoted by $t_{\text{L13}}^{\text{C}}$.

We take the solid mantle bounded on top by a TMO of 100, 500 and 1000 km thickness, and use Eq. (15) to determine the timescales of half-equilibration as a function of phase change number, $\Phi^+$. In an attempt to apply our fitting equation to

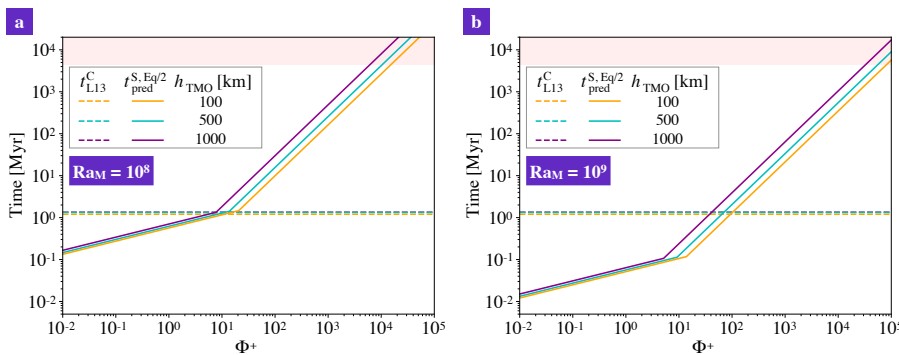

**Figure 7.** Timescales to reach chemical half-equilibrium between the solid mantle and a TMO of 100, 500 and 1000 km (respectively orange, blue and purple solid lines, this study, $t_{\text{pred}}^{\text{S, Eq/2}}$), and timescales of crystallisation of that TMO until it is completely mushy (thicknesses with same colours in dashed lines, from Lebrun et al. (2013), $t_{\text{L13}}^{\text{C}}$), versus different values of phase change number, $\Phi^+$. a) corresponds to a Rayleigh number of the solid part of $\text{Ra}_{\text{M}} = 10^8$ and b) to $\text{Ra}_{\text{M}} = 10^9$. Rose background corresponds to a time higher than the age of the Earth. We note that $t_{\text{L13}}^{\text{C}}$ are consistent with the timescales constrained by Nikolaou et al. (2019) and Salvador et al. (2017).

Earth, we assume that the global Rayleigh number of the early-Earth mantle just after solidification of the TMO is between
$\text{Ra}_{\text{M}} = 10^8$ and $\text{Ra}_{\text{M}} = 10^9$. $\text{Ra}_{\text{M}}$ is calculated on the basis of the total thickness of the solid mantle, $h_{\text{M}}$. The Rayleigh number, Ra, used in Eq. (15) is then re-scaled to the actual thickness of the solid mantle (i.e., before solidification of the TMO) as follows:

$$\text{Ra} = \text{Ra}_{\text{M}} \left( \frac{h_{\text{S}}}{h_{\text{M}}} \right)^3. \tag{16}$$

This re-scaling neglects the change of various physical parameters (from Eq. (6)), but is sufficient for our discussion.

The comparison between timescales is presented in Fig. 7. The timescale to crystallise the TMO is loosely dependent on its thickness, and this time is around 1 Myr. Our models predict that there are significant chemical exchanges between the TMO and the solid mantle for $\Phi^+ <10$ and $<100$, for $\text{Ra}_{\text{M}} = 10^8$ and $\text{Ra}_{\text{M}} = 10^9$, respectively. Therefore, for $\Phi^+$ smaller than these values, the TMO is expected to have reached (at least) chemical half-equilibrium with much of the mantle before reaching the mush stage. Therefore, a very strong enrichment of the final-stage TMO as predicted by fractional crystallisation models (Elkins-Tanton et al., 2003) is not expected to occur for small-to-moderate $\Phi^+$.

Increasing the thickness of the TMO, hence, decreasing the thickness of the solid mantle, decreases the Rayleigh number and the ratio $V_{\text{S}}/V_{\text{M}}$, which make the dimensionless time increase (see fitting). But also, decreasing the thickness of the solid mantle decreases the scale for time ($h_{\text{S}}^2/\kappa$), which partially compensates the increase mentioned previously when recovering the dimensional time. As a result, the thickness of solid in the phase-change regime only loosely affects the dimensional half-equilibration time (Fig. 7). The effect of the thickness of solid is a bit stronger in the high-$\Phi$ regime (as mentioned before).

## 4 Discussion

Our models address the compositional evolution of the solid mantle bounded by magma oceans above and/or below, and constrain the time needed to chemically (half-)equilibrate these reservoirs. While the concept of a TMO that potentially interacts with the underlying solid mantle is now well accepted, the idea of a long-lived BMO remains controversial. Whether or not a BMO can be stabilised depends on the slope of the adiabat vs. that of the melting curve (Labrosse et al., 2007), and/or on the fate of FeO-rich TMO cumulates that sink to the CMB (Labrosse et al., 2015; Ballmer et al., 2017b). Regardless of these issues, our study can be applied to various scenarios, including those with a solid mantle bounded just by a TMO, and bounded by a TMO and BMO.

Classical fractional crystallisation models predict strongly inverse chemical stratification of the initial solid mantle, and consequently a global-scale overturn by the end of a TMO crystallisation (e.g., Elkins-Tanton et al., 2003, 2005). The propensity of such a massive density-driven overturn depends on whether or not chemical equilibration between the solid mantle and magma ocean(s) can occur before magma-ocean solidification. Any style of magma-ocean crystallisation leaves a strongly super-adiabatic thermal profile, which should drive convection in the cumulate layers before full solidification (Solomatov and Stevenson, 1993a; Ballmer et al., 2017b; Maurice et al., 2017; Boukaré et al., 2018), and related vertical flow should promote (partial) melting near the TMO-mantle (and BMO-mantle) boundary/ies. In this study we focus on a phase change boundary condition that allows material to flow through the boundary/ies and continuously change the composition of solid and liquid reservoirs. While flow through the boundary/ies remains penalised by the phase-change number, $\Phi$, this approach implies final degrees of melting of at least $\sim$40%, i.e. beyond the rheological transition, at which crystals become suspended (Abe, 1997; Costa et al., 2009). Future work using a more realistic melting model is needed to test whether these high degrees of melting can indeed be reached, or quantify the effects of partial melting on equilibration between the mantle and TMO/BMO.

Our models show that the composition of the solid mantle and magma oceans strongly depends on the phase change number, $\Phi$. In this study we take $\Phi$ as being constant through time, but because this number depends on the dynamics and thicknesses of the magma oceans, $\Phi$ may change continuously in a more realistic model with moving boundaries. Although we show that chemical equilibration can occur before full crystallisation of the magma oceans, variations of $\Phi$ and a moving-boundary scheme should be considered in further studies.

Considering that $\Phi^{\pm}$ values are low when TMO and BMO (or just TMO) crystallisation starts (Morison et al., 2019; Morison, 2019), mantle convection would first assume a degree-1 pattern (Fig. 5), possibly with implications for the origin of crustal dichotomy on the Moon (e.g., Ishihara et al., 2009) and Mars (e.g., Roberts and Zhong, 2006; Citron et al., 2018). However, it remains to be shown that such a degree-1 pattern of convection would be able to survive through all stages of magma-ocean crystallisation.

The crystallisation fronts move at different speeds, since the TMO can crystallise in a few Myr years (Lebrun et al., 2013; Salvador et al., 2017; Nikolaou et al., 2019), whereas the BMO may persist for much longer (e.g., Labrosse et al., 2007, 2015). Therefore, $\Phi^{\pm}$ would change accordingly. The efficiency of equilibration during the late-stage magma ocean depends on the timescale of freezing of this final stage, as well as on the efficiency of mass transfer ($\Phi^{+}$) for a thin and partially mushy TMO.

Once the TMO is fully crystallised, $\Phi^+$ tends to infinity, while $\Phi^-$ assumes a finite value as long as the BMO is still present. Dynamics in the solid mantle would change accordingly: convection in the solid mantle may be either dominated by degree-1 (low $\Phi^-$) or by higher degrees of convection (high $\Phi^-$) (Fig. 5). Although our models do not account for core cooling explicitly, the heat transfer across the mantle is expected to be much more efficient for lower values of $\Phi^-$ than for high values. This implies that the BMO is likely to crystallise much faster than suggested by Labrosse et al. (2007, 2015) for low $\Phi^-$, at

least as long as no dense FeO-enriched materials accumulate at the bottom of the solid mantle to prevent efficient mass transfer across the BMO-mantle boundary. The BMO may even be thermally coupled to the relatively fast-cooling TMO for low $\Phi^+$ and low $\Phi^-$.

On the other hand, it is conceivable that thermally-coupled TMO and BMO crystallise more slowly than expected for a thermally-isolated TMO (Agrusta et al., 2019). The presence of a BMO makes heat transfer across the mantle and out of

the thermally coupled BMO and core more efficient than for cases without a BMO and with a boundary layer at the CMB instead. Such a situation implies that there is a larger heat reservoir available to buffer the temperature of the TMO for a given heat flux through the atmosphere and to space. Note that the mass and heat capacity of the core are similar to that of a 1000 km-thick magma ocean, hence the timescale for full crystallisation could be roughly twice than usually computed (cf. Lebrun et al. (2013)). However, the timescales of TMO, BMO and core cooling would be largely unaffected if the BMO were thermo-

chemically stratified (Laneuville et al., 2018). Whether or not material transfer across the whole mantle, as predicted here for cases with low $\Phi^\pm$, can efficiently cool the core has important implications for the long-term thermal evolution of terrestrial planets, and the propensity of an (early) dynamo.

Even though timescales of BMO crystallisation are not well constrained, chemical exchange between the two magma oceans (through the solid mantle) is still likely to occur. Note that the same process (i.e. mantle convection) that takes out heat from

the BMO and core, is responsible for this chemical exchange. As an example for the Earth, if we take a TMO and a BMO of 100 km thickness each, $\Phi^\pm \leq 10$ and a Rayleigh number of $10^8$, we would expect a half-equilibrium between solid mantle, TMO and BMO in less than $\sim 460$ ky, i.e. before TMO crystallisation (and even more so, before BMO crystallisation). This chemical exchange, however, does not necessarily imply homogeneity between the TMO and BMO, because the relevant phase diagrams that control fractional crystallisation at the TMO-solid mantle (low pressures) and BMO-solid mantle (high

pressures) boundaries are very distinct (e.g., Thomas et al., 2012; Boukaré et al., 2015). For example, while the FeO distribution coefficient, $K$ defined in Eq. (11), is taken as constant in this study, its value is likely to be pressure-dependent (Nomura et al., 2011; Miyazaki and Korenaga, 2019a), potentially causing partitioning of FeO into the BMO. Regardless, any such exchange between the TMO, solid mantle and BMO could be a way to sequester trace elements (including heat-producing elements) into the BMO, particularly if the TMO freezes faster than the BMO.

Once both oceans crystallise and $\Phi^\pm \to \infty$, convection in the solid mantle likely changes to higher degrees of convection (as already seen with $\Phi = 10^2, 10^3$ in Fig. 4b and Fig. 4c), similar to present-day Earth-mantle dynamics. Our models predict a largely homogeneous solid mantle, with some regions preserving significant primordial heterogeneity for long timescales. The preservation of heterogeneity is likely to be enhanced once composition-dependent rheology (i.e., a difference in intrinsic strength of mantle materials) is considered (Manga, 1996; Ballmer et al., 2017a; Gülcher et al., 2020). Indeed, primordial

cumulates formed in the lower mantle may be strongly enriched in MgSiO$_3$ bridgmanite (Boukaré et al., 2015), and hence intrinsically strong (Yamazaki and Karato, 2001). In the present-day, LLSVPs are perhaps the most prominent and seismically-evident large-scale mantle heterogeneities. That they are only rather mildly Fe-enriched (Deschamps et al., 2012) points to rather efficient equilibration between the magma ocean(s) and much of the solid mantle during crystallisation, such as predicted by a subset of our models. For the Earth, the subset of our models with $\Phi^+$ smaller than $\sim$100 suggests that chemical (half-

)equilibrium between a solid mantle and a (100 to 1000 km-thick)TMO can be accomplished in less than $\sim 1$ Myr, i.e., before the TMO is fully solidified or becomes a mush (Fig. 7). Equilibration in such sort timescale makes the solid mantle mostly homogeneous (with some heterogeneities as seen before), which could explain the pyrolitic nature of the mantle (Wang et al., 2015; Zhang et al., 2016; Kurnosov et al., 2017). Therefore, the final-stage TMO and subsequent mush may be efficiently equilibrated with most of the solid mantle. In this case, we expect solid compositions that are by far not as enriched in FeO as

predicted by fractional crystallisation models (e.g. Elkins-Tanton, 2012), in which strong enrichment only occurs because the final-stage TMO is fully separated from the solid mantle, with strong disequilibrium between the two reservoirs.

Similarly, we expect moderate enrichment (in FeO and incompatible trace elements) and roughly basaltic-to-komatiitic (i.e., the melting product of a hot $\sim$pyrolitic mantle) major-element compositions of the primary crust. As our models do not explicitly address the final and mush stages of the TMO, and consider only a strongly simplified compositional model with

400 only two components, (Fe, Mg)O, more detailed studies with a more complex compositional treatment are needed in order to predict the composition of the early crust.

In our models, we consider a simplified initial condition with bulk-planetary TMO and BMO compositions ($X_{\text{FeO}}^{\text{Bulk}} = 0.12$ and $X_{\text{FeO}}^{\text{TMO, BMO}} = 0.12$). While this condition may be realistic for a formation of the solid mantle due to equilibrium crystalli-sation, the TMO and BMO would be significantly more FeO enriched initially if they were formed by fractional crystallisation

(Solomatov and Stevenson, 1993a; Xie et al., 2020). In our models, the initial cumulate downwellings formed at the TMO-solid mantle boundary are depleted in FeO and hence buoyant, resisting solid-mantle convection and delaying compositional equi-libration, but this effect would be strongly diminished, or even opposite, for a more realistic initial condition. Conversely, the initially depleted cumulate upwellings from the BMO-mantle boundary in our models advance convection and equilibration. As these effects that depend on our choice of the initial condition scale with buoyancy number, B, we choose a conservative

value of B = 1.0 (see Section 2.1). Using higher values of B (B $\approx$ 3) is expected to advance TMO-mantle equilibration for fractional crystallisation of the solid mantle shell. Our models predict that such an equilibration can occur swiftly to avoid extreme enrichments of the TMO during progressive crystallisation, and thus to prevent a subsequent global-scale overturn with deep-mantle stratification. In our setup, the impact of the value of B is fairly limited due to weak compositional contrasts. Indeed, Fig. 4 exhibits compositional contrasts of at most $\Delta X_{\text{FeO}}^{\text{S}} = 0.08$. With B $\approx$ 3, the buoyancy would still be dominated

by the thermal term of order $\Delta T = 1$, rather than by compositional buoyancy B$\Delta X_{\text{FeO}}^{\text{S}} \approx 0.3$. The value of B will only have a significant impact in the late stages of the crystallisation of the magma ocean, when fractional crystallisation leads to strong compositional contrasts.

Smaller planets than Earth are less likely to be chemically equilibrated for a given bulk composition. First, they tend to cool faster, as they contain a smaller total reservoir of heat and volatiles (i.e., stabilising a less massive atmosphere to shield

cooling). Moreover, the Ra number is lower for small planets, such that equilibration is expected to take longer according to our results. Thus, the Martian mantle might be less equilibrated (more stratified) than that of Earth (Elkins-Tanton et al., 2003; Maurice et al., 2017). On the other hand, Super-Earths are expected to be well equilibrated, particularly as BMOs are likely to be stabilised in their interiors due to high CMB pressures (Stixrude, 2014; Caracas et al., 2019), which has a strong effect on equilibration timescales. Whether or not chemical equilibration during the magma-ocean stage is efficient has important

implications for the composition of the primary crust, the propensity of overturn and related stabilisation of a deep dense layer, as well as the long-term evolution of terrestrial planets.

## 5 Conclusions

In this work we use a numerical model to investigate the thermo-chemical evolution of the convecting solid mantle bound at top and/or bottom by magma oceans. We parameterise fractional crystallisation and melting processes of dynamic topography at

430 either or both solid mantle boundaries, and determine the timescales to reach chemical half-equilibrium between solid mantle and magma ocean(s).

We show that these fractional crystallisation and dynamic melting processes at either or both boundaries play an important role in the chemical evolution of the solid mantle. Efficient transfer of FeO across the mantle-TMO and/or mantle-BMO boundary can prevent strong enrichment of the last-stage magma ocean, and thereby any strong chemical stratification of the

435 early fully-solid mantle. Moreover, this efficient transfer of FeO renders the timescales of chemical (half-)equilibration between the solid mantle and magma ocean(s) shorter than (or on the order of) 1 Myr. Since magma ocean crystallisation occurs in few Myr (Abe, 1997; Lebrun et al., 2013; Salvador et al., 2017; Nikolaou et al., 2019), our study suggests that chemical equilibrium between solid and liquid reservoirs can be reached before the end of magma ocean crystallisation. Therefore, a strong chemical stratification of the solid mantle is unlikely to occur, and the first crust is not expected to be extremely enriched in FeO.

This prediction fundamentally contrasts with that of classical models of fractional crystallisation of the magma ocean (e.g. Elkins-Tanton, 2012).

However, more studies are needed to better constrain chemical-equilibration timescales. This could be achieved, for instance, as more realistic compositional models and phase diagrams are accounted for, and/or a moving boundary approach is applied to explicitly model the evolution of either or both crystallisation fronts.

## 445 Appendix A: Regression Method

The best fitting coefficients of all regressions are obtained using a simple algorithm. Each free parameter has an initial possible minimum and maximum, chosen here between -1 and 1. All 8 parameters $a_i$ in Eq. 15 are scanned between these minimum and maximum boundaries using homogeneous steps. For each point in that 8-D space, we compute the misfit between predicted and observed timescale as $\log \sum_{\text{cases}} \left( t_{\text{pred}}^{\text{S,Eq/2}} - t_{\text{case}}^{\text{S,Eq/2}} \right)^2$. The set of best fitting parameters are found by selecting the lowest

misfit between the analytical formulation and the data.

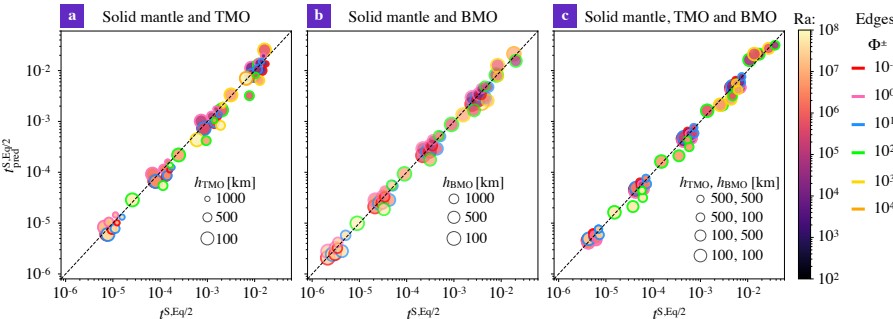

**Figure A1.** Regression for all data with Eq. (15), for the solid mantle bound by a) a TMO of 1000, 500 and 100 km ($V_S = 4.7 \times 10^{20}, 6.7 \times 10^{20}$ and $8.6 \times 10^{20}$ m$^3$, respectively), b) a BMO of 1000, 500 and 100 km ($V_S = 7.1 \times 10^{20}, 8.2 \times 10^{20}$ and $8.9 \times 10^{20}$ m$^3$, respectively), c) and a TMO and BMO of 500 - 500 km, 500 - 100 km, 100 - 500 km, and 100 - 100 km ($V_S = 5.8 \times 10^{20}, 6.6 \times 10^{20}, 7.7 \times 10^{20}$, and $8.4 \times 10^{20}$ m$^3$, respectively). Colours indicate the corresponding Rayleigh number, Ra, and colour edges represent the phase change number, $\Phi$.

When the best fitting coefficients are found after a first search, new iterations of the algorithm are requested using more refined windows in the parameter space located around the previous best fitting parameters. When a best parameter is found at the boundary of the parameter space, the parameter space is widened such that the best fitting coefficients are independent from the initial boundaries in parameter space. Iterations of the search are performed until the solution is converged below fourth

digit precision.

Figure A1 shows the regression for all cases that reached chemical half-equilibrium.

*Author contributions.* D. P. Bolrão, M. D. Ballmer, A. Morison, A. B. Rozel, S. Labrosse and P. J. Tackley designed the study. D. P. Bolrão, A. Morison, A. B. Rozel, P. Sanan, S. Labrosse and P. J. Tackley developed the code. M. D. Ballmer, A. Morison, A. B. Rozel and S. Labrosse supported D. P. Bolrão in investigating the results. A. B. Rozel fitted the data and obtained the empirical scaling law. D. P. Bolrão made the

figures and wrote the paper draft. All co-authors provided input and suggestions for the paper draft.

*Competing interests.* The authors declare that they have no conflict of interest.

*Acknowledgements.* We thank Dr. Antonio Manjón-Cabeza Córdoba, the Editor Julien Aubert and two anonymous reviewers for useful comments that improved the first version of this manuscript. We gratefully acknowledge support from the SNSF grant 200021E-164337 and ANR-15-CE31-0018-01.

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
