# Peer review of "Timescales of chemical equilibrium between the convecting solid mantle and over-/underlying magma oceans"

_Solid Earth, 2020_

## Referee Comment (RC1) · Anonymous Referee #1 · 26 May 2020

**General comments**

Bolrão *et al.* use the model of phase-change (permeable) dynamical boundary conditions (BCs) applied to a solid mantle bounded by a top magma ocean (TMO) and/or a basal magma ocean (BMO) to compute the timescale for chemical equilibration between these different reservoirs. The phase-change BCs, initially derived to describe the inner-core boundary, have been recently introduced in the context of magma oceans, and have been shown to possibly induce strong differences with cases considering classical free-slip BCs. This work takes a step forward in the task of constraining the effects of this dynamical setting in a realistic context. Specifically, it presents the

first results of finite-amplitude thermo-chemical convection simulations using this type of BCs in a cylindrical geometry, using a spherical annulus mesh.

Depending on the "permeability" of the boundaries, quantified by the phase-change number (noted $\Phi$), as well as the geometry of the problem (presence and thickness of a BMO and/or a TMO), the authors show that the convection pattern associated with the phase-change BCs (already presented in previous studies, e.g. Agrusta *et al.*, 2019) are recovered in this setting, and that they strongly influence the timing of chemical equilibration between the solid cumulates and the magma ocean(s). On the one hand, when the efficiency of phase-change at the boundary is low, mass transfer between the reservoirs is limited and chemical equilibration is portracted (or even never reached). On the other hand, when mass transfer is efficient at the boundarie(s), chemical equilibration is significantly sped up, and can be achieved on timescales shorter than those associated with magma ocean crystallization. The later case would rule out strong density heterogeneity in the solid cumulates induced by fractional crystallization of the magma ocean(s), and the resulting scenario of mantle overturn. The authors derive a scaling law to compute the characteristic time for equilibration as a function of $\Phi$, the Rayleigh number, and the volume fraction of the already solidified mantle, that can be readily used by the community.

The article reads generally well, makes clear points and draws meaningful conclusions. I think that the description of the numerical setting and the fitting process should be elaborated, and the choice of figures adapted (see the specific comments). My main concern is about the treatment of chemical buoyancy. While the results on chemical equilibration are relevant for all sorts of species affected by fractionation (volatiles, heat-producing elements, trace elements etc.), the framework of this work is that of Fe fractionation, which has an impact on the dynamics of the system by inducing density anomalies. This is accounted for in the model. However, the effects of compositional buoyancy on the flow are not discussed. Furthermore, the density difference between the two compositional end-members considered being probably uncertain (the twocomponent model itself being a simplification), I would have expected this density difference (i.e. the buoyancy number) to be one of the parameters of the study. Yet only one value is considered, and is not even motivated. Actually, doing a quick calculation with $\alpha = 2 \times 10^{-5}\text{K}^{-1}$, $\Delta T$=2000 K (typical value for the geometry used here with the melting curves from Fiquet *et al.*, 2010 for the initial temperature profile), $\rho_{\text{mantle}} = 4000$ kg/m$^3$ (as in Ballmer *et al.*, 2017) and B=1 (the value used in the present work), I found: $\Delta\rho = \alpha\Delta T B\rho_{\text{mantle}} = 160$ kg/m$^3$, which is about one order of magnitude lower than what you would expect for pure FeO and pure MgO end-members (e.g. Boukaré *et al.*, 2015). Therefore I think the authors should either more strongly motivate their choice of B=1, or consider testing several values for it. For instance, using B=0, they could extend their discussion to strictly passively advected material, like trace elements.

**Specific comments**

- Lines 37 and 46: I am a bit confused here: the opening and concluding sentences "the solidus is steeper than the isentrope" and "the adiabat is steeper than the melting curve" seem contradictory. If you do mean the that the adiabat is steeper than the melting curve (which you need for re-melting of sinking, Fe-rich cumulates), it seems to me that you are already in the middle-out crystallization case. Or do you expect the adiabat to be steeper than the melting curves only in the solid mantle?

- Figure 1: Although it is made clear that the curvature of the liquidus curve in panel b is exaggerated, I am a bit puzzled by the fact that the temperature decreases in the bottom of the mantle, rather than only increasing at a lower rate than the adiabats. I don't think anyone predicts that the temperature of the melting curves actually decrease with depth (it just increases at a lower rate than the adiabat).

- I do not understand if the computing mesh changes with the geometry of the
case: although it is suggested in Figure 2 (with varying $R^+$ and $R^-$), I don't really see it in Figure 5 (but maybe the outer-to-inner radius differences between those cases are too small, in which case that might be notified in the caption).

- Please, include a table with the values of the different parameters and quantities used: $h_{\text{TMO}}$, $h_S$, $h_{\text{BMO}}$ (and related $R^+/R^-$ if relevant), Ra (and/or SC and Ra$_c$), B, $\Phi^\pm$, K, $X_{\text{Fe}}^{\text{bulk}}$.

- Line 86-87: "We ensure mechanical stability between the solid mantle and magma oceans, i.e., $\rho_{\text{TMO}} < \rho_S < \rho_{\text{BMO}}$". How do you do that? As far as I understand, density is only parametrized by $X_{\text{Fe}}$, and when you reach equilibrium, both TMO and BMO have the same $X_{\text{Fe}}$ which should imply: $\rho_{\text{TMO}} = \rho_{\text{BMO}}$. But anyway the density of the magma oceans is not considered in this study (there is no other reference to $\rho_{\text{MO}}$ in the text except in Fig. 2), so this sentence might be superfluous.

- Lines 143-146: This fact is important and would deserve attention (in future studies). The melt/freeze boundary conditions have been developed to study the inner core boundary where a unique melting temperature can be defined. For mantle rock, as pointed out in the text, the temperature span between solidus and liquidus probably induces different behavior, which is hard to tell *a priori*.

- You assume that at equilibrium, $X_{\text{Fe}}$ in the solid is homogeneous, but I can imagine that overturn of heavy cumulates could result in a layered configuration and an associated layered convection pattern where FeO would be sequestered at the bottom, resulting in a Fe-rich BMO, a Fe-poor TMO, and heterogeneous (layered) mantle.

- A few more words about how particles are handled would be welcome. For generalities (e.g. advection algorithm), references to previous work would be sufficient, but I guess new techniques were introduced for this study, whose description

could benefit to the community. In particular, how do you ensure the mass conservation with permeable boundaries: do you balance the number of particles going out at the "melting" interface with that coming in at the "freezing" one? And how do you distribute the incoming particles?

- Figure 4: Decimals in non-dimensional time are superfluous. Moreover, since the convection is mainly thermal, having snapshots of the temperature could help, especially for the case with a low value of $\Phi$, since it is an unusual convection pattern. If you do, you might consider discarding some timesteps which are not so important to understand the evolution, in order save space on the figure.

- The half-equilibrium time is parameterized using the parameters of the study: Ra, $\Phi$ and $V_S/V_M$. It would be interesting to discuss what might be the influence of other parameters that where not varied in this study (e.g. the buoyancy number, the partition coefficient or the bulk $X_{\mathsf{Fe}}$).

- Figure 7 is hard to read, and largely redundant with Figure 6. One important new information is that increasing $\Phi$ increases the influence of the volume of the solid mantle, but it is already mentioned in the text. If the point is to represent the good agreement of the scaling law with the predictions, I think Figure A1 is sufficient.

- Several more recent studies on the timescale for crystallization of a terrestrial magma ocean have been published since Lebrun *et al.* 2013:

    – Salvador et al., The relative influence of $H_2O$ and $CO_2$ on the primitive surface conditions and evolution of rocky planets, JGR: Planets 122, 2017.
    – Nikolaou et al., What factors affect the duration and outgassing of the terrestrial magma ocean? ApJ 875, 2019.

- In the discussion you suggest that $\Phi$ is low when the crystallization starts (line 309), and that $\Phi <\sim 100$ is a "realistic value" (line 382), but there is no discussion

on the expected evolution of $\Phi$, so you should at least cite some previous studies where it is explained.

- I don't really understand the fitting algorithm: Do you scan all parameters at once, or do you fit them one after the other? Do you choose which branch of the scaling law (i.e. which set of parameters) is fitted depending on the location in the parameter space (i.e. implying the regime boundary)? Since it is an appendix, I think you might develop this (very succinct) description, or even write the algorithm as pseudo-code if it is not too long. It would be also a good place to define what you call "error" in Table 1.

**Technical corrections**

- Line 26: "crystals start to appear and consolidate..."

- Line 42: ".. become denser with time." You could refer to Figure 1a where this process is represented.

- Line 58: As for Line 42, I would also refer to Figure 1b.

- Line 118: "is noted $\tau_\eta$" rather than "is given by $\tau_\eta$".

- Line 160-161: Depending on what you mean, I would rather write that "FeO and MgO are thought to be the Fe-rich and Mg-rich end-members of mantle silicates" or that "FeO and MgO represent the Fe-rich and Mg-rich end-members of mantle silicates".

- Line 172: "(similar to a half-life)" I would introduce the notion of half-equilibrium after calculating the equilibrium.

- Line 194: I think the correct word here is "bounded" (as you use it further).

- Line 208: "... in dimensionless time units..."

- Line 210: "... thereby bringing the solid mantle and the TMO close to chemical equilibrium"

- Line 213: since you're giving the half-equilibrium times in non-dimensional units, which are not very insightful, it might be better to compare these times between each other (e.g. saying that half equilibrium is reached for $\Phi$=10-1 ~10 times faster than for $\Phi = 10^2$, and ~200 faster than for $\Phi = 10^3$).

- Line 214: "... for these three cases." or "... for these three values of $\Phi^-$

- Figures 4 and 5: What sets the streamlines' color-code? Maybe having them just white would avoid confusing with FeO content in the cases where mixing induces small-scale heterogeneites.

- Line 249: Shouldn't it be a minimum rather than a maximum?

- Caption Figure 6: "white circles" instead of "white colours".

- Table 1: It is not clear to me what the "error" is in this context.

- Line 257: "Our models predict that in the regime of efficient material transfer (i.e., for low values of $\Phi$), timescales to reach chemical half-equilibrium are virtually unaffected by the volume of the solid mantle" I would then expect $a_3$ to be close to 0, why is it not the case?

- Line 270: "Agrusta *et al.*, 2019 showed"

- Line 303: Whether or not chemical equilibration occurs between the solid mantle and magma ocean(s) is highly influential on the extent of this initial chemical stratification."

• Line 328-329: "Note that the thermal inertia of the core is similar to ..."

---

## Referee Comment (RC2) · Anonymous Referee #2 · 3 Jun 2020

This paper provides a numerical model of solid mantle convection below, above or sandwiched between magma ocean(s) with phase change boundaries. The latter allows material transport between the solid mantle and the magma ocean and is used instead of the commonly used free-slip boundary condition. At the boundary between a solid mantle and a liquid of similar composition, a flow through the phase change can take place, depending on how fast latent heat is transferred in the liquid region. Due to convection in the solid mantle a dynamic topography is generated which can be eroded by melting around topographic highs and is 'filled' by freezing, around topography depression. This process requires that the latent heat released in regions of freezing is transferred efficiently to regions where it is consumed for melting. The

critical parameter for the efficiency of material exchange is the phase change number Phi, the ratio between phase-change timescale for transferring latent heat from region where it is released to places where it is consumed and the timescale for producing a dynamic topography. The authors find that a small value of the phase change boundary (Phi <100) allows efficient chemical equilibration before the end of magma ocean crystallization of about 1 Ma even in case of fractional crystallization. This material exchange may prevent strong chemical stratification of the solid mantle and enforce chemical homogenization – a finding in contrast to classical models of fractional crystallization. This is an interesting model and expands on previous work of this group related to the interaction between solid mantle and top or basal magma oceans. In the present study, the focus is on the timescales of chemical equilibrium between the solid mantle and magma ocean(s). Similar to previous studies on the interaction between the solid mantle and the magma ocean, the most unknown but important parameter is Phi, and its value is highly speculative because the model is in part very simple and important physical processes may not be captured - most of them were mentioned on paper. For example, for silicate material there is no sharp boundary between liquid and solid phase. The partial melting zone could be responsible for inefficient heat transport of latent heat and may lead to significantly higher Phi. This heat transport is possibly even more inefficient if the phase boundary moves (solid mantle grows) through cooling. The authors therefore investigate a large range of Phi, which is certainly positive, but unfortunately does not make it easier to classify the results. However, at this stage one has to accept this uncertainty, but there are some aspects of this work that could be better explained and/or discussed in more detail.

Points that should be considered / discussed:

Specific comments

1. Thermo-chemical convection: The number of buoyancy is set to 1 without further explanation. Why was this value taken and how does the choice of B influence the results? With the present values it seems that the chemical density variation has no

significant influence.

2. In the initial setup a homogeneous FeO content of solid mantle and magma ocean is assumed. As the authors themselves write, this is not a realistic initial state but it also not clear how sensible are the obtained time scales depending on initial conditions? In the current setup, the material that forms in the topography depression is depleted in FeO, this would not be the case for a more realistic start condition. An initial unstable gradient in the solid mantle can trigger convection but may also result is a stable configuration after overturn, depending on B (see above). This can be important for the time scale of chemical equilibrium - if in this case a chemical equilibrium can be established.

3. Two effects have been neglected, but they can also result in chemical equilibration and compositional mixing before final magma ocean crystallization: 1) When the solid mantle grows and has no fixed boundaries, as is assumed here, convection causes the new top crystallized layer, which should have a different FeO content, to sink and mix continuously with the solid mantle. 2) If convection in the solid mantle starts before the solidification of MO, partial melting of the cumulates and 'feeding' of the MO with this melt is very likely. Both effects change the chemical equilibrium considerably and do not necessarily require the material to be able to flow through by phase change. However, the latter may further reduce the time scale of chemical equilibrium.

4. Steady-state simulations, i.e. delta T is constant, but also no internal heat sources and a constant viscosity are used – all these effects can influence the strength of convection (and chemical equilibration) and possibly the convection pattern. In particular the influence of internal heat and a temperature dependent viscosity could be tested fast.

5. A new phase change boundary condition for convection has been investigated, are there benchmark studies also for sufficient resolution or how do the authors ensure that the calculations are correct?
6. A table should be added for all parameters used.

7. Line 255: It is not clear to me why with small Phi the volume of the solid mantle has no effect on the time scale of chemical equilibration. Do the authors have an explanation? I think this also indicates that the material flow is extremely large - is this really realistic? One could estimate the value.

minor comments

8. Line 310: In the discussion, the crustal dichotomies of Mars and the Moon are mentioned and associated with the present process. I don't find this so obvious, because according to the model low degree convection is postulated at the beginning of the MO crystallization, but the crustal dichotomy is more likely to occur at the end of the MO phase, when the pattern becomes small scale.

9. Line 370: It is stated that smaller planets cool faster. This is not generally true, for example if a blanketing crust is formed during MO crystallization before the mantle is entirely solid and the cooling and crystallization of the MO slows down considerably - as postulated for the Moon.

10. Line 380: "for realistic values for the phase change number Phi+ smaller than $\sim$ 100". I doubt that we really know the realistic value in view of the simplification of the process and the unknown parameters.

technical comment

11. Figure 7 is difficult to read with the different symbols and lines.

---

## Editor Comment (EC1) · Julien Aubert (Editor) · 4 Jun 2020

Dear authors,

at this stage you have received two referee comments on your submission. Both referees acknowledge the interest of this work, but between them, they raise a number of important point which should all be addressed before we can go further.

Please revise your manuscript accordingly and post your responses to the referees comments within the interactive discussion.

Julien Aubert

---

## Author Comment (AC1) · 10 Jul 2020

July 10, 2020

**Response to the Reviewers and Editor**

We would like to thank the Editor for handling our paper and the Reviewers for taking the time to review our manuscript so thoughtfully. We took the useful comments and pertinent questions into consideration, which have substantially improved our paper. Both reviewers address an important point regarding the value of the Buoyancy number, B. We agree that the value used in our simulations (1.0) is lower than a realistic value (3.0 or more), and we decided to show results for a more reasonable value of B. Therefore, at the moment we are trying to run new simulations with a proper value for B. Unfortunately, we don't have the new results yet since our clusters are still facing problems from a cyber attack that occurred late-May. We plan to have the results in the next couple of months.

Therefore, we would like to ask if it possible to have an extension of two months, to address the B problem properly. The rest of our responses are reported below.

**Response to Reviewer Number 1**

**General comments**

**Reviewer:** My main concern is about the treatment of chemical buoyancy. While the results on chemical equilibration are relevant for all sorts of species affected by fractionation (volatiles, heat-producing elements, trace elements etc.), the framework of this work is that of Fe fractionation, which has an impact on the dynamics of the system by inducing density anomalies. This is accounted for in the model. However, the effects of compositional buoyancy on the flow are not discussed. Furthermore, the density difference between the two compositional end-members considered being probably uncertain (the two-component model itself being a simplification), I would have expected this density difference (i.e. the buoyancy number) to be one of the parameters of the study. Yet only one value is considered, and is not even motivated. Actually, doing a quick calculation with $\alpha = 2 \times 10^{-5}$ K$^{-1}$, $\Delta T = 2000$ K (typical value for the geometry used here with the melting curves from Fiquet et al., 2010 for the initial temperature profile), $\rho_{mantle} = 4000$ kg/m$^3$ (as in Ballmer et al., 2017) and B=1 (the value used in the present work), I found: $\Delta \rho = \alpha \Delta T B \rho_{mantle} = 160$ kg/m$^3$, which is about one order of magnitude lower than what you would expect for pure FeO and pure MgO end-members (e.g. Boukaré et al., 2015). Therefore I think the authors should either more strongly motivate their choice of B=1, or consider testing several values for it. For instance, using B=0, they could extend their discussion to strictly passively advected material, like trace elements.

→ **Authors:** Indeed the value of B is lower than expected for Earth. We did not vary the value of B because our parameter space was already too vast. We decided to keep B = 1.0 because our preliminary results showed that B does not have a huge impact on the dynamics of the solid mantle. However, we agree that in order to avoid confusion is probably better to show results for a more reasonable value of B. Therefore, at the moment we are trying

to run new simulations with B = 3.0 (this value comes from $B = \frac{\Delta\rho_{ch}}{\alpha\Delta T\rho_0} = \frac{670}{0.00002\times3000\times4000} = 2.8$). Unfortunately, we don't have the new results yet. Our clusters are still facing problems from a cyber attack that occurred late-May. We plan to have the results in the next couple of months.

**Specific comments**

**Reviewer:** Lines 37 and 46: I am a bit confused here: the opening and concluding sentences "the solidus is steeper than the isentrope" and "the adiabat is steeper than the melting curve" seem contradictory. If you do mean the that the adiabat is steeper than the melting curve (which you need for re-melting of sinking, Fe-rich cumulates), it seems to me that you are already in the middle-out crystallization case. Or do you expect the adiabat to be steeper than the melting curves only in the solid mantle?

→ **Authors:** We decided to change the sentence in the text.

**Reviewer:** Figure 1: Although it is made clear that the curvature of the liquidus curve in panel b is exaggerated, I am a bit puzzled by the fact that the temperature decreases in the bottom of the mantle, rather than only increasing at a lower rate than the adiabats. I don't think anyone predicts that the temperature of the melting curves actually decrease with depth (it just increases at a lower rate than the adiabat).

→ **Authors:** We agree it is not the best representation. It seemed to us that in order to draw a basal magma ocean, we would have to exaggerate the curvature of the liquidus. We put more thought into this figure and we changed

it in a way that the temperature of the liquidus increases at a lower rate than the adiabats. The new figure is now included in the paper.

**Reviewer:** I do not understand if the computing mesh changes with the geometry of the case: although it is suggested in Figure 2 (with varying R+ and R-), I don't really see it in Figure 5 (but maybe the outer-to-inner radius differences between those cases are too small, in which case that might be notified in the caption).

→ **Authors:** Meshes are indeed different, however, differences between the aspect ratio of case are too small to be noticed (differences are only obvious if one makes use of a ruler). We agree that this is a point that should be addressed to avoid confusion. We modified the caption of figure 5 and added this note.

**Reviewer:** Please, include a table with the values of the different parameters and quantities used: hTMO, hS, hBMO (and related R+/R- if relevant), Ra (and/or SC and Rac), B, $\Phi^{\pm}$, K, $X_{bulk}^{Fe}$.

→ **Authors:** Included.

**Reviewer:** Line 86-87: "We ensure mechanical stability between the solid mantle and magma oceans, i.e., $\rho_{TMO} < \rho_S < \rho_{BMO}$". How do you do that? As far as I understand, density is only parametrized by XFe, and when you reach equilibrium, both TMO and BMO have the same XFe which should imply: $\rho_{TMO} = \rho_{BMO}$. But anyway the density of the magma oceans is not considered in this study (there is no other reference to $\rho_{MO}$ in the text except in Fig. 2), so this sentence might be superfluous.

→ **Authors:** It is a good point. This is just a misuse of the word "ensure" from our part. We assume this mechanical stability condition holds true

(and physically, it is necessary for $\Phi$ to be positive). We changed the word "ensure" to "assume" for clarity.

**Reviewer:** Lines 143-146: This fact is important and would deserve attention (in future studies). The melt/freeze boundary conditions have been developed to study the inner core boundary where a unique melting temperature can be defined. For mantle rock, as pointed out in the text, the temperature span between solidus and liquidus probably induces different behavior, which is hard to tell a priori.

→ **Authors:** It's a good point and indeed this should be taken into account in future projects!

**Reviewer:** You assume that at equilibrium, XFe in the solid is homogeneous, but I can imagine that overturn of heavy cumulates could result in a layered configuration and an associated layered convection pattern where FeO would be sequestered at the bottom, resulting in a Fe-rich BMO, a Fe-poor TMO, and heterogeneous (layered) mantle.

→ **Authors:** We do not assume that XFe in the solid is homogeneous. We show that although average composition of the solid mantle tend to mutual chemical equilibrium, chemical homogeneity across the solid mantle is not necessarily reached (figure 4). As the reviewer points out, the solid mantle ends up strongly heterogeneous (albeit without layering), and the implications are discussed in the paper. Instead, the timescale of "half-equilibration" provides a measure of significant chemical changes in the magma ocean relative to the *average* mantle. We now explain this more clearly in the text. The thermo-chemical evolution anticipated by the reviewer (in terms of layering) is not predicted by any of our models (and again nothing is imposed that would prevent such an evolution). In a current study (in prep.), we explore the effects of initial mantle stratification in the

solid (while the TMO/BMO are still present), but the related consequences go beyond the scope of this paper.

**Reviewer:** A few more words about how particles are handled would be welcome. For generalities (e.g. advection algorithm), references to previous work would be sufficient, but I guess new techniques were introduced for this study, whose description could benefit to the community. In particular, how do you ensure the mass conservation with permeable boundaries: do you balance the number of particles going out at the "melting" interface with that coming in at the "freezing" one? And how do you distribute the incoming particles?

→ **Authors:** This deserves attention indeed. We changed the explanation in "2.3 Compositional treatment" to address this point. We simulate melting of solid material when dynamic topography develops outside the solid domain, i.e., when there is an outflux of material of the solid domain. Melting is simulated assuming that no fractionation operates when the solid melts, i.e., all (Fe,Mg)O present in this topography goes into the magma ocean. Therefore, tracers that leave the domain pass their information (about mass and composition) to the magma ocean, and are deleted. We simulate crystallisation of the magma ocean when negative dynamic topography develops in the solid domain. When this happens, cells near the boundary are left with no tracers, so new tracers have to be introduced in those cells. These new tracers simulate solid being created. We calculate the influx of mass corresponding to this dynamic topography, and distribute this mass to the new tracers. The composition of the solid created is related to that of the liquid by fractional crystallisation. Therefore, only a fraction of FeO goes into the solid, and this fraction is given by the partition coefficient, $K$.

**Reviewer:** Figure 4: Decimals in non-dimensional time are superfluous. Moreover, since the convection is mainly thermal, having snapshots of the temperature

could help, especially for the case with a low value of $\Phi$, since it is an unusual convection pattern. If you do, you might consider discarding some timesteps which are not so important to understand the evolution, in order save space on the figure.

→ **Authors:** Regarding "Decimals in non-dimensional time are superfluous", we agree and modified the figure to reduce the number of decimals. Regarding the temperature snapshots: we will add them once we have new results.

**Reviewer:** The half-equilibrium time is parameterized using the parameters of the study: Ra, $\Phi$ and VS/VM. It would be interesting to discuss what might be the influence of other parameters that where not varied in this study (e.g. the buoyancy number, the partition coefficient or the bulk XFe).

→ **Authors:** As mentioned previously in this document, simulations at $B = 3$ are ongoing. The partition coefficient is not expected to affect results tremendously when varying in its fairly narrow range of realistic values. It is harder to predict the effect of the bulk XFe, and in particular how it is distributed between the solid and the ocean(s) in the initial condition. We chose here to focus on the consequences of the phase change boundary condition to show it is an important ingredient when considering the chemical evolution of our system. We do agree that it would be important to study the effect of the initial condition, but we think it is out of scope of this paper, and better suited in more realistic studies about the long term evolution of the solid/magma ocean(s) system.

**Reviewer:** Figure 7 is hard to read, and largely redundant with Figure 6. One important new information is that increasing $\Phi$ increases the influence of the volume of the solid mantle, but it is already mentioned in the text. If the point is to represent

the good agreement of the scaling law with the predictions, I think Figure A1 is sufficient.

→ **Authors:** We agree that figure 7 may be too complicated due to the amount of information. We remove figure 7 from this manuscript, since the main information is written in the text, and the good agreement of the scaling law with the predictions is made in figure A1.

**Reviewer:** Several more recent studies on the timescale for crystallization of a terrestrial magma ocean have been published since Lebrun et al. 2013: – Salvador et al., The relative influence of H2O and CO2 on the primitive surface conditions and evolution of rocky planets, JGR: Planets 122, 2017. – Nikolaou et al., What factors affect the duration and outgassing of the terrestrial magma ocean? ApJ 875, 2019.

→ **Authors:** Unfortunately these citations were missing in the first version of this manuscript. We acknowledge the work made by Salvador et al. and Nikolaou et al. and correct the mistake in this new version of the paper.

**Reviewer:** In the discussion you suggest that $\Phi$ is low when the crystallization starts (line 309), and that $\Phi \sim 100$ is a "realistic value" (line 382), but there is no discussion on the expected evolution of $\Phi$, so you should at least cite some previous studies where it is explained.

→ **Authors:** We make a note in the methodology section that Morison et al. (2019) and Morison (2019) estimated $\Phi^+ \sim 10^{-5}$ and $\Phi^- \sim 10^{-3}$, for a purely thermal case. So small values of $\Phi$ are expected for the Earth. However, our paper includes a compositional treatment, and given that $\Phi^{\pm}$ is difficult to constrain, we vary this value between $10^{-1}$ and $10^5$ (we use $\Phi^{\pm} = 10^{-1}$ as the lowest value possible for $\Phi^{\pm}$ because the resolution of the

thermal boundary layer is computationally demanding once $\Phi^{\pm}$ decreases below $10^{-1}$). Later in the conclusion we mention $\Phi \sim 100$ is a "realistic value" and this may be confusing indeed. What we meant is that low values of $\Phi$ are expected (Morison et al. (2019) and Morison (2019)), and we show that at any value below $\Phi^{\pm} \sim 100$, the system reaches chemical (half-)equilibration before magma ocean crystallisation (previous figure 8, now figure 7). We changed this sentence to "Moreover, this efficient transfer of FeO renders the timescales of chemical (half-)equilibration between the solid mantle and magma ocean(s) shorter than (or on the order of) 1 Myr."

**Reviewer:** I don't really understand the fitting algorithm: Do you scan all parameters at once, or do you fit them one after the other? Do you choose which branch of the scaling law (i.e. which set of parameters) is fitted depending on the location in the parameter space (i.e. implying the regime boundary)? Since it is an appendix, I think you might develop this (very succinct) description, or even write the algorithm as pseudo-code if it is not too long. It would be also a good place to define what you call "error" in Table 1.

→ **Authors:** This is a good point and we explain it better in the appendix now. All parameters are scanned at the same time until one solution is found. Once the solution is found the window search of each parameter is redefined to values closer to the value found. This process is done multiple times, to refine the solution.

**Technical corrections**

**Reviewer:** • Line 26: "crystals start to appear and consolidate..." • Line 42: ".. become denser with time." You could refer to Figure 1a where this process is

represented. â˘Aΰ Line 58: As for Line 42, I would also refer to Figure 1b. â˘Aΰ Line 118: "is noted $\tau_\eta$" rather than "is given by $\tau_\eta$". â˘Aΰ Line 160-161: Depending on what you mean, I would rather write that "FeO and MgO are thought to be the Fe-rich and Mg-rich end-members of mantle silicates" or that "FeO and MgO represent the Fe-rich and Mg-rich end-members of mantle silicates". â˘Aΰ Line 172: "(similar to a half-life)" I would introduce the notion of half-equilibrium after calculating the equilibrium. â˘Aΰ Line 194: I think the correct word here is "bounded" (as you use it further). â˘Aΰ Line 208: ". . . in dimensionless time units..." â˘Aΰ Line 210: ". . . thereby bringing the solid mantle and the TMO close to chemical equilibrium" â˘Aΰ Line 213: since you're giving the half-equilibrium times in non-dimensional units, which are not very insightful, it might be better to compare these times between each other (e.g. saying that half equilibrium is reached for $\Phi = 10^{-1} \sim 10$ times faster than for $\Phi = 10^2$, and $\sim 200$ faster than for $\Phi = 10^3$). â˘Aΰ Line 214: "... for these three cases." or "... for these three values of $\Phi^-$ â˘Aΰ Figures 4 and 5: What sets the streamlines' color-code? Maybe having them just white would avoid confusing with FeO content in the cases where mixing induces small-scale heterogeneites. â˘Aΰ Line 249: Shouldn't it be a minimum rather than a maximum? â˘Aΰ Caption Figure 6: "white circles" instead of "white colours". â˘Aΰ Table 1: It is not clear to me what the "error" is in this context. â˘Aΰ Line 257: "Our models predict that in the regime of efficient material transfer (i.e., for low values of $\Phi$), timescales to reach chemical half-equilibrium are virtually unaffected by the volume of the solid mantle" I would then expect a3 to be close to 0, why is it not the case? â˘Aΰ Line 270: "Agrusta et al., 2019 showed" â˘Aΰ Line 303: Whether or not chemical equilibration occurs between the solid mantle and magma ocean(s) is highly influential on the extent of this initial chemical stratification." C7 â˘Aΰ Line 328-329: "Note that the thermal inertia of the core is similar to ..."

→ **Authors:** All these comments are very useful and we tried to address them all in the text.

**Response to Reviewer Number 2**

**Specific comments**

**Reviewer Point 1:** Thermo-chemical convection: The number of buoyancy is set to 1 without further explanation. Why was this value taken and how does the choice of B influence the results? With the present values it seems that the chemical density variation has no significant influence.

→ **Authors:** Indeed the value of B we use is lower than what is expected for Earth. We did not vary the value of B because our parameter space was already too vast. However, we agree that it is probably better to show results for a more reasonable value of B. Therefore, at the moment we are trying to run new simulations with B = 3.0 (this value comes from $B = \frac{\Delta\rho_{ch}}{\alpha\Delta T\rho_0} = \frac{670}{0.00002\times3000\times4000} = 2.8$). Unfortunately, we don't have the new results yet. Our clusters are still facing problems from a cyber attack that occurred late-May. We plan to have the results in the next couple of months.

**Reviewer Point 2:** In the initial setup a homogeneous FeO content of solid mantle and magma ocean is assumed. As the authors themselves write, this is not a realistic initial state but it also not clear how sensible are the obtained time scales depending on initial conditions? In the current setup, the material that forms in the topography depression is depleted in FeO, this would not be the case for a more realistic start condition. An initial unstable gradient in the solid mantle can trigger convection but may also result is a stable configuration after overturn, depending on B (see above). This can be important for the time scale of chemical equilibrium - if in this case a chemical equilibrium can be established.

→ **Authors:** This is a good point. In the new simulations that are currently

running with $B = 3$, we also use a more realistic initial condition which leads to material enriched in FeO in newly formed solid. Overall, it is difficult to fully assess how the initial condition would affect results, especially as this makes the parameter space much larger. We chose here to focus on the consequences of the phase change boundary condition to show it is an important ingredient when considering the chemical evolution of our system. We do agree that it would be important to study the effect of the initial condition, but we think it is out of scope of this paper, and better suited in more realistic studies about the long term evolution of the solid/magma ocean(s) system.

**Reviewer Point 3:** Two effects have been neglected, but they can also result in chemical equilibration and compositional mixing before final magma ocean crystallization: 1) When the solid mantle grows and has no fixed boundaries, as is assumed here, convection causes the new top crystallized layer, which should have a different FeO content, to sink and mix continuously with the solid mantle. 2) If convection in the solid mantle starts before the solidification of MO, partial melting of the cumulates and 'feeding' of the MO with this melt is very likely. Both effects change the chemical equilibrium considerably and do not necessarily require the material to be able to flow through by phase change. However, the latter may further reduce the time scale of chemical equilibrium.

→ **Authors:** These are two important points and we modified the manuscript to address them. Regarding the first one: in this paper we assume fixed boundaries and test different thicknesses of the solid mantle (or magma oceans). Indeed a moving boundary would create a new top crystallised layer, which should have a different FeO content, and sink and mix continuously with the solid mantle. This effect could result in chemical equilibration and compositional mixing before final magma ocean crystallisation. However, we show here that with the phase change boundary condition, chemical equilibration

can occur on a timescale that is much smaller than that of crystallization. Integrating the effect of the moving boundary is much more demanding computationally (an evolution model for magma oceans is necessary, as well as some way to deal with moving boundaries). This will be the subject of more complete studies in the future. Note also that in this study we take $\Phi$ as being constant through time, but because this number depends on the dynamics and thicknesses of the magma oceans, $\Phi$ may change continuously in a more realistic model with moving boundaries. Variations of $\Phi$ and a moving-boundary scheme should definitely be considered in further studies to study the long-term evolution of the solid/magma ocean(s) system.

Regarding point 2: in this study we focus our attention to a phase change boundary condition, that allows material to flow through the boundary and continuously change the composition of solid and liquid reservoirs. Partial melting of solid cumulates can indeed still change the composition of the magma ocean, without the use of this boundary condition. However, we show that with this boundary condition a larger volume of material can (re-)melt and crystallise efficiently at either or both solid-liquid phase boundaries.

**Reviewer Point 4:** Steady-state simulations, i.e. delta T is constant, but also no internal heat sources and a constant viscosity are used – all these effects can influence the strength of convection (and chemical equilibration) and possibly the convection pattern. In particular the influence of internal heat and a temperature dependent viscosity could be tested fast.

→ **Authors:** Indeed these effects can influence the results. But testing the effect of internal heat sources, as well as temperature dependent viscosity, goes beyond the scope of the current paper. These should be definitely taken into account in future projects.

**Reviewer Point 5:** A new phase change boundary condition for convection has been investigated, are there benchmark studies also for sufficient resolution or how do the authors ensure that the calculations are correct?

→ **Authors:** The phase change boundary condition implementation was tested against linear stability analysis (Agrusta et al., 2019 for cartesian geometry; Morison, 2019 spherical geometry). In this study we also check energy conservation over the solid mantle, and iron mass conservation over the whole mantle (solid mantle and magma oceans).

**Reviewer Point 6:** A table should be added for all parameters used.

→ **Authors:** Included.

**Reviewer Point 7:** Line 255: It is not clear to me why with small Phi the volume of the solid mantle has no effect on the time scale of chemical equilibration. Do the authors have an explanation? I think this also indicates that the material flow is extremely large - is this really realistic? One could estimate the value.

→ **Authors:** Coefficients of the fitting equation indicate that at low values of $\Phi$, the ratio between volumes (i.e., the aspect ratio of the evolution scenario), has less impact than at high values of $\Phi$. One possible explanation for this is that at low values of $\Phi$, convection occurs with low degree, so the geometry of the problem is less important. We address this in the new version of the manuscript. Extremely large material flow is realistic, since $\Phi^+ \sim 10^{-5}$ and $\Phi^- \sim 10^{-3}$, according to Morison et al. (2019) and Morison (2019), for a purely thermal case.

**Reviewer Point 8:** Line 310: In the discussion, the crustal dichotomies of Mars and the Moon are mentioned and associated with the present process. I don't find this so

obvious, because according to the model low degree convection is postulated at
the beginning of the MO crystallization, but the crustal dichotomy is more likely
to occur at the end of the MO phase, when the pattern becomes small scale.

→ **Authors:** We modified the text to address this point.

**Reviewer Point 9:** Line 370: It is stated that smaller planets cool faster. This is not gen-
erally true, for example if a blanketing crust is formed during MO crystallization
before the mantle is entirely solid and the cooling and crystallization of the MO
slows down considerably - as postulated for the Moon.

→ **Authors:** This is a good point and we modified the text to address it. Indeed
it is not clear what could happen to smaller planets than Earth. In these
smaller planets, Ra number is lower, which could imply that equilibration
may take longer. However, it also depends on the timescale of crystallisation
of the magma ocean. On one hand, smaller planets tend to cool faster, as
they contain a smaller total reservoir of heat (and volatiles), but on the other
hand, if a blanketing crust is formed before the mantle is entirely crystallised,
then the magma ocean would take longer to cool.

**Reviewer Point 10:** Line 380: "for realistic values for the phase change number Phi+
smaller than $\sim 100$". I doubt that we really know the realistic value in view of the
simplification of the process and the unknown parameters.

→ **Authors:** Indeed we don't know an exact value of $\Phi$. However, Morison et
al. (2019) and Morison (2019) estimated $\Phi^+ \sim 10^{-5}$ and $\Phi^- \sim 10^{-3}$, for
a purely thermal case. So small values of $\Phi$ are expected for the Earth.
However, our paper includes a compositional treatment, and given that $\Phi^\pm$
is difficult to constrain, we vary this value between $10^{-1}$ and $10^5$ (we use
$\Phi^\pm = 10^{-1}$ as the lowest value possible for $\Phi^\pm$ because the resolution of the

thermal boundary layer is computationally demanding once $\Phi^{\pm}$ decreases below $10^{-1}$). In this paper we compare our estimated timescales for chemical (half-)equilibration with timescales of magma ocean crystallisation given by Lebrun et al., 2013. We see that for values of $\Phi$ lower that $\sim 100$, chemical (half-)equilibration occurs before magma ocean crystallisation (previous figure 8, now figure 7). Indeed the sentence "for realistic values for the phase change number Phi+ smaller than $\sim$ 100" might lead to confusion. What we meant is that low values of $\Phi$ are expected (Morison et al. (2019) and Morison (2019)), and we show that at any value below $\Phi^{\pm} \sim 100$, the system reaches chemical (half-)equilibration before magma ocean crystallisation. We changed this sentence to "Moreover, this efficient transfer of FeO renders the timescales of chemical (half-)equilibration between the solid mantle and magma ocean(s) shorter than (or on the order of) 1 Myr."

**Reviewer Point 11:** Figure 7 is difficult to read with the different symbols and lines.

$\rightarrow$ **Authors:** We agree that figure 7 may be too complicated due to the amount of information. We remove figure 7 from this manuscript, since the main information is written in the text, and the good agreement of the scaling law with the predictions is made in figure A1.

---

## Referee Report (RR1)

**Second review for: "Timescales of chemical equilibrium between the convecting solid mantle and over-/underlying magma oceans" by Bolrão *et al*.**

**General comments**

While most points I raised about presentation and explanations have been satisfactorily taken into consideration, my concern on the effect of chemical buoyancy could not be addressed due to technical issues that I can appreciate. This is unfortunate because the article would clearly benefit from including the buoyancy number in its parameter space. However the results on the timescale of equilibration between mantle and magma ocean(s) are still valuable, but the discussion should elaborate more on the the composition of the magma ocean (see specific comments). The paragraph which has been added about the effect of chemical buoyancy is not entirely satisfactory in my opinion, and could point out the limitation of having B=1 rather than trying to convince the reader that the effect of this parameter is not important, relying on (what I found) a shaky argument. Finally, I am also puzzled by the use of the term "chemical equilibrium", which, in its common understanding, is something very different from what is dealt with here. I would suggest "mass" or "composition" equilibrium, which doesn't have as strong an underlying meaning.

**Specific comments**

L 46-47: *Such overturn(s) may lead to re-melting of FeO-enriched material at depth, as the isentrope of such material is steeper than its melting curve through most of the mantle.*

This needs to be supported by a citation. Actually in Labrosse et al., 2015, this mechanism is indeed suggested to produce Fe-rich melts, but not because the isentrope of the Fe-rich material is steeper than the melting curve (which, incidentally, would hinder thermal overturn). Possible re-melting is rather attributed to viscous heating of material already close to their solidus.

L 110-113: *Regarding the buoyancy number, although earth like models point to a value of B∼3.0 today, the value in the early Earth might be different. In this paper we make a conservative choice and consider a buoyancy number of B = 1.0, to attribute similar weight to compositional and thermal effects on the density.*

Here a short discussion on what B could be in the early Earth (at least whether it might be higher or lower) would be relevant. Besides, I don't really understand in which sense the choice of B=1 is "conservative". A value higher than the expected one (2.8 based a the present-day Earth mantle) would be conservative in the sense that one would be sure not to underestimate the effect of chemical buoyancy.

Furthermore, since the $X^S_{FeO}$ cannot reach high values (in particular it cannot be higher than $X^{MO}_{FeO,}$ which is lower than ~0.25 in the cases presented in Figure 3), the available density contrast in the mantle is reduced, so taking B=1 attributes an effective weight to compositional effects that is significantly lower than the thermal ones (the total temperature range being ensured to be covered due to the boundary conditions).

L 177: *We calculate the influx of mass corresponding to this dynamic topography, and distribute this mass by the new tracers*

How do you relate the number of tracers to the inflowing mass? Based on their initial density (i.e. number of tracers per cell)? Or do you balance the number of tracers lost at melting boundaries to that of tracers gained at crystallizing boundaries?

L 208-287:

In parts 3.1 and 3.2 you present results in terms of time for half equilibrium for cases having both a 2390-km-thick mantle (either TMO or BMO) and a 2290-km-thick mantle (both BMO and TMO). The time normalization is different for those two cases (one non-dimensional unit time for the former is about 9% longer than one non-dimensional unit time for the later). Since you compare results from both cases and use them to fit the scaling law, they would probably need to be normalized to a common scale; are they? Since the result exhibits variations in orders of magnitude for the tie of half equilibrium, I don't expect the conclusion to change.

L 396 *but this effect would be strongly diminished for a more realistic initial condition.*
This effect could even be reversed if the TMO is sufficiently enriched compared to the mantle.

L 400-402: *Thus, the actual value considered for B is expected to have only a minor effect on the equilibration timescales constrained here.*

I am not entirely convinced by this reasoning. What you observe is that, for B=1, magma ocean(s) and mantle equilibrate swiftly, diminishing the influence of chemical buoyancy. However, in order to conclude that the effect of B is minor, you have to assume that equilibration between magma ocean(s) and mantle will be fast regardless of the value of B, which cannot be justified based on the present work.

Here are a couple of ideas to extend the discussion:

Your results seem to suggest that the magma ocean crystallizes in the equilibrium crystallization regime (where it is in equilibrium with the whole mantle at all times). In this case, you could easily compute the evolution of $X^{MO}_{FeO}$ and the crystallizing material's $X_{FeO}$ as a function of the magma ocean's volume and be more quantitative about the magma ocean's enrichment. Moreover, you could do the same for lower partition coefficient and extend the discussion to other species (e.g. volatiles: how do your results compare with the assumptions for volatiles partitioning from Lebrun *et al.*, 2013 and the other studies?).

I would also suggest, for the available simulations, to study the time evolution of $X_{FeO}$ in the crystallizing material (I guess it can be easily derived from the value of $X^{MO}_{FeO}$) and to compare it to the average value in the mantle. This could help discussing the effective chemical density contrast.

**Technical corrections**

L 271-272: *One possible explanation for this is that at low values of Φ, convection occurs with low degree, so the geometry of the problem is less important.*

Is it a "possible explanation" for the previous statement ("this conclusion is independent of the evolution scenario") or for the weak influence of the solid mantle's volume?

L 349: *it is conceivable that a thermally-coupled TMO and BMO crystallise more slowly than expected*

it is conceivable that  thermally-coupled TMO and BMO crystallise more slowly than expected

L 394: *(i.e., a more realistic initial condition at least for the TMO)*
I don't understand what you mean here.

---

## Author Response (AR2)

**Timescales of chemical equilibrium between the convecting solid mantle and over-/underlying magma oceans**

Daniela Bolrão, Maxim Ballmer, Adrien Morison,
Antoine Rozel, Patrick Sanan, Stéphane Labrosse, Paul Tackley

October 23, 2020

**Response to the Reviewers and Editor**

We would like to thank the Editor for handling our paper and the Reviewers for taking the time to review our manuscript so thoughtfully. We took the useful comments and pertinent questions into consideration, which have substantially improved our paper. Both reviewers address an important point regarding the value of the Buoyancy number, B. We agree that the value used in our simulations (B = 1.0) is lower than a realistic/expected value (B ≥ 3.0), and we agreed that it would be good to show results for a higher value of B. Therefore, we decided to re-run all simulations with a more suitable value of B. We would like to thank the Editor and the Editorial Board once again for kindly granting us extensions of the submission deadline of this paper, so we could show the new results. However, unfortunately our simulations are not ready and we apologise for this matter.

Our simulations took longer to run due to a cyber attack that occurred late-May, which prevented our clusters to work normally and effectively in the past months. Moreover, we noticed that few simulations that reached the end are quantitatively wrong, due to a small bug in the code. Although this is just a quantitative error, results cannot be published. However, the new preliminary results look similar to the results presented in this paper. We are convinced that the value of B doesn't bring major changes in the timescales of chemical equilibrium between the solid mantle and magma oceans. Nevertheless, in this version of the paper we defend the point that B = 1.0 is a conservative choice and by using it, it gives a similar weight to compositional and thermal effects on the density (lines 110-113). The implications for planetary evolution are still important and relevant (lines 391-403).

Once again, we would like to thank the Editor, the Reviewers and the Editorial Board for handling our paper and apologise for the delay in the simulations. Please find our responses to the comments of the Reviewers below.
* * *
**Response to Reviewer Number 1**
**General comments**

**Reviewer:** My main concern is about the treatment of chemical buoyancy. While the results on chemical equilibration are relevant for all sorts of species affected by fractionation (volatiles,

heat-producing elements, trace elements etc.), the framework of this work is that of Fe fractionation, which has an impact on the dynamics of the system by inducing density anomalies. This is accounted for in the model. However, the effects of compositional buoyancy on the flow are not discussed. Furthermore, the density difference between the two compositional end-members considered being probably uncertain (the two-component model itself being a simplification), I would have expected this density difference (i.e. the buoyancy number) to be one of the parameters of the study. Yet only one value is considered, and is not even motivated. Actually, doing a quick calculation with $\alpha = 2 \times 10^{-5}$ K$^{-1}$, $\Delta T = 2000$ K (typical value for the geometry used here with the melting curves from Fiquet et al., 2010 for the initial temperature profile), $\rho_{mantle} = 4000$ kg/m$^3$ (as in Ballmer et al., 2017) and B=1 (the value used in the present work), I found: $\Delta\rho = \alpha \Delta T B \rho_{mantle} = 160$ kg/m$^3$, which is about one order of magnitude lower than what you would expect for pure FeO and pure MgO end-members (e.g. Boukaré et al., 2015). Therefore I think the authors should either more strongly motivate their choice of B=1, or consider testing several values for it. For instance, using B=0, they could extend their discussion to strictly passively advected material, like trace elements.

$\rightarrow$ **Authors:** Indeed the value of B is lower than expected for the Earth today. Today, that value would be around B = 2.8 (this value comes from B = $\frac{\Delta\rho_{ch}}{\alpha\Delta T\rho_0} = \frac{670}{2\times10^{-5}\times3000\times4000} =$ 2.8), but in an early Earth this value may be different. In this paper we take a conservative choice of B by assuming B = 1.0, which attributes a similar weight to compositional and thermal effects on the density. We address this point in the methods section (lines 110-113) and discussion section (lines 391-403).

**Specific comments**

**Reviewer:** Lines 37 and 46: I am a bit confused here: the opening and concluding sentences "the solidus is steeper than the isentrope" and "the adiabat is steeper than the melting curve" seem contradictory. If you do mean the that the adiabat is steeper than the melting curve (which you need for re-melting of sinking, Fe-rich cumulates), it seems to me that you are already in the middle-out crystallization case. Or do you expect the adiabat to be steeper than the melting curves only in the solid mantle?

$\rightarrow$ **Authors:** We changed the sentence in the text for clarification (lines 37-38 and 46-47).

**Reviewer:** Figure 1: Although it is made clear that the curvature of the liquidus curve in panel b is exaggerated, I am a bit puzzled by the fact that the temperature decreases in the bottom of the mantle, rather than only increasing at a lower rate than the adiabats. I don't think anyone predicts that the temperature of the melting curves actually decrease with depth (it just increases at a lower rate than the adiabat).

$\rightarrow$ **Authors:** We agree it is not the best representation. We put more thought into this figure and we changed it in a way that the temperature of the liquidus increases at a lower rate than the adiabats. The new figure is now included in the paper (Fig. 1).

**Reviewer:** I do not understand if the computing mesh changes with the geometry of the case: although it is suggested in Figure 2 (with varying R+ and R-), I don't really see it in Figure 5 (but maybe the outer-to-inner radius differences between those cases are too small, in which case that might be notified in the caption).

→ **Authors:** Meshes are indeed different, however, differences between the aspect ratio of case are too small to be noticed (differences are only obvious if one makes use of a ruler). We agree that this is a point that should be addressed to avoid confusion. We modified the caption of Fig. 5 and added this note.

**Reviewer:** Please, include a table with the values of the different parameters and quantities used: hTMO, hS, hBMO (and related R+/R- if relevant), Ra (and/or SC and Rac), B, $\Phi^{\pm}$, K, $X_{bulk}^{Fe}$.

→ **Authors:** It is now included (Table 1).

**Reviewer:** Line 86-87: "We ensure mechanical stability between the solid mantle and magma oceans, i.e., $\rho_{TMO} < \rho_S < \rho_{BMO}$". How do you do that? As far as I understand, density is only parametrized by XFe, and when you reach equilibrium, both TMO and BMO have the same XFe which should imply: $\rho_{TMO} = \rho_{BMO}$. But anyway the density of the magma oceans is not considered in this study (there is no other reference to $\rho_{MO}$ in the text except in Fig. 2), so this sentence might be superfluous.

→ **Authors:** This is a good point. This is just a misuse of the word "ensure" from our part. We assume this mechanical stability condition holds true (and physically, it is necessary for $\Phi$ to be positive). We changed the word "ensure" to "assume" for clarity (line 89). That this assumption is valid is related to the liquid-solid density relationships as are sketched in Figure 1b.

**Reviewer:** Lines 143-146: This fact is important and would deserve attention (in future studies). The melt/freeze boundary conditions have been developed to study the inner core boundary where a unique melting temperature can be defined. For mantle rock, as pointed out in the text, the temperature span between solidus and liquidus probably induces different behavior, which is hard to tell a priori.

→ **Authors:** It's a good point and indeed this should be taken into account in future projects.

**Reviewer:** You assume that at equilibrium, XFe in the solid is homogeneous, but I can imagine that overturn of heavy cumulates could result in a layered configuration and an associated layered convection pattern where FeO would be sequestered at the bottom, resulting in a Fe-rich BMO, a Fe-poor TMO, and heterogeneous (layered) mantle.

→ **Authors:** We do not assume that XFe in the solid is homogeneous. Our models predict that the system tends to chemical equilibrium between the (average) solid mantle and the magma oceans (lines 221-224 and Fig. 3). That said, in many cases there is still significant heterogeneity across the solid mantle (as can be observed in Fig. 4 and

Fig. 5, and clarified in lines 234-238), but the average composition is in ∼equilibrium with the over/underlying liquid. This is a model prediction, not an assumption. The idea of the reviewer that the BMO becomes progressively FeO enriched is not predicted by any of our models. Due to continuous melting/crystallization at the BMO-mantle boundary, the BMO also tends to chemical equilibration with the mantle.

**Reviewer:** A few more words about how particles are handled would be welcome. For generalities (e.g. advection algorithm), references to previous work would be sufficient, but I guess new techniques were introduced for this study, whose description could benefit to the community. In particular, how do you ensure the mass conservation with permeable boundaries: do you balance the number of particles going out at the "melting" interface with that coming in at the "freezing" one? And how do you distribute the incoming particles?

→ **Authors:** This deserves attention indeed. We amended the explanation in "2.3 Compositional treatment" to address this point.

**Reviewer:** Figure 4: Decimals in non-dimensional time are superfluous. Moreover, since the convection is mainly thermal, having snapshots of the temperature could help, especially for the case with a low value of $\Phi$, since it is an unusual convection pattern. If you do, you might consider discarding some timesteps which are not so important to understand the evolution, in order save space on the figure.

→ **Authors:** Regarding "Decimals in non-dimensional time are superfluous", we agree and modified the figure to reduce the number of decimals. Regarding the temperature: we tried to include temperature snapshots but the figure got extremely complicated. However, temperature field follows the pattern of the composition (same contour lines).

**Reviewer:** The half-equilibrium time is parameterized using the parameters of the study: Ra, $\Phi$ and VS/VM. It would be interesting to discuss what might be the influence of other parameters that where not varied in this study (e.g. the buoyancy number, the partition coefficient or the bulk XFe).

→ **Authors:** The partition coefficient is not expected to affect results tremendously when varying in its fairly narrow range of realistic values (we added a brief discussion in lines 366) It is harder to predict the effect of the bulk XFe, and in particular how it is distributed between the solid and the ocean(s) in the initial condition. We chose here to focus on the consequences of the phase change boundary condition to show it is an important ingredient when considering the chemical evolution of our system. We do agree that it would be important to study the effect of the initial condition, but we think it is out of scope of this paper, and better suited in more realistic studies about the long term evolution of the solid/magma ocean(s) system.

**Reviewer:** Figure 7 is hard to read, and largely redundant with Figure 6. One important new information is that increasing $\Phi$ increases the influence of the volume of the solid mantle, but it is already mentioned in the text. If the point is to represent the good agreement of the scaling law with the predictions, I think Figure A1 is sufficient.

→ **Authors:** We agree that figure 7 may be too complicated due to the amount of information. We remove figure 7 from this manuscript, since the main information is written in the text, and the good agreement of the scaling law with the predictions is made in figure A1.

**Reviewer:** Several more recent studies on the timescale for crystallization of a terrestrial magma ocean have been published since Lebrun et al. 2013: – Salvador et al., The relative influence of H2O and CO2 on the primitive surface conditions and evolution of rocky planets, JGR: Planets 122, 2017. – Nikolaou et al., What factors affect the duration and outgassing of the terrestrial magma ocean? ApJ 875, 2019.

→ **Authors:** We added these references. In Figure 7 caption, we also note now that the crystallisation timescales constrained by Lebrun et al., Nikolaou et al., and Salvador et al. are consistent with each other (which is particularly true on the log-scale of Figure 7).

**Reviewer:** In the discussion you suggest that $\Phi$ is low when the crystallization starts (line 309), and that $\Phi \sim 100$ is a "realistic value" (line 382), but there is no discussion on the expected evolution of $\Phi$, so you should at least cite some previous studies where it is explained.

→ **Authors:** There is a detailed introduction/discussion on the relevance of "realisic"/expected values of phi, i.e. low values (see method section "2.2 Dynamic topography and the phase change boundary condition"). We reworded somewhat misleading statements in the discussion section and added some citations of previous studies.

**Reviewer:** I don't really understand the fitting algorithm: Do you scan all parameters at once, or do you fit them one after the other? Do you choose which branch of the scaling law (i.e. which set of parameters) is fitted depending on the location in the parameter space (i.e. implying the regime boundary)? Since it is an appendix, I think you might develop this (very succinct) description, or even write the algorithm as pseudo-code if it is not too long. It would be also a good place to define what you call "error" in Table 1.

→ **Authors:** This is a good point and we improved the explanation of the fitting procedure in the Appendix.

**Technical corrections**

**Reviewer:** • Line 26: "crystals start to appear and consolidate..." • Line 42: ".. become denser with time." You could refer to Figure 1a where this process is represented. • Line 58: As for Line 42, I would also refer to Figure 1b. • Line 118: "is noted $\tau_\eta$" rather than "is given by $\tau_\eta$". • Line 160-161: Depending on what you mean, I would rather write that "FeO and MgO are thought to be the Fe-rich and Mg-rich end-members of mantle silicates" or that "FeO and MgO represent the Fe-rich and Mg-rich end-members of mantle silicates". • Line 172: "(similar to a half-life)" I would introduce the notion of half-equilibrium after calculating the equilibrium. • Line 194: I think the correct word here is "bounded" (as you use it further). • Line 208: ". . . in dimensionless time units..." • Line 210: ". . .

thereby bringing the solid mantle and the TMO close to chemical equilibrium" • Line 213: since you're giving the half-equilibrium times in non-dimensional units, which are not very insightful, it might be better to compare these times between each other (e.g. saying that half equilibrium is reached for $\Phi = 10^{-1} \sim 10$ times faster than for $\Phi = 10^2$, and $\sim 200$ faster than for $\Phi = 10^3$). • Line 214: "... for these three cases." or "... for these three values of $\Phi^-$ • Figures 4 and 5: What sets the streamlines' color-code? Maybe having them just white would avoid confusing with FeO content in the cases where mixing induces small-scale heterogeneites. • Line 249: Shouldn't it be a minimum rather than a maximum? • Caption Figure 6: "white circles" instead of "white colours". • Table 1: It is not clear to me what the "error" is in this context. • Line 257: "Our models predict that in the regime of efficient material transfer (i.e., for low values of $\Phi$), timescales to reach chemical half-equilibrium are virtually unaffected by the volume of the solid mantle" I would then expect a3 to be close to 0, why is it not the case? • Line 270: "Agrusta et al., 2019 showed" • Line 303: Whether or not chemical equilibration occurs between the solid mantle and magma ocean(s) is highly influential on the extent of this initial chemical stratification." C7 • Line 328-329: "Note that the thermal inertia of the core is similar to ..."

→ **Authors:** All these comments are very useful and we tried to address them all in the text.
* * *
**Response to Reviewer Number 2**
**Specific comments**

**Reviewer Point 1:** Thermo-chemical convection: The number of buoyancy is set to 1 without further explanation. Why was this value taken and how does the choice of B influence the results? With the present values it seems that the chemical density variation has no significant influence.

→ **Authors:** In this paper we take a conservative choice of B by assuming B = 1.0, which attributes a similar weight to compositional and thermal effects on the density (lines 110-113). We added few lines in the discussion that describe briefly the influence of B (lines 391-403).

**Reviewer Point 2:** In the initial setup a homogeneous FeO content of solid mantle and magma ocean is assumed. As the authors themselves write, this is not a realistic initial state but it also not clear how sensible are the obtained time scales depending on initial conditions? In the current setup, the material that forms in the topography depression is depleted in FeO, this would not be the case for a more realistic start condition. An initial unstable gradient in the solid mantle can trigger convection but may also result is a stable configuration after overturn, depending on B (see above). This can be important for the time scale of chemical equilibrium - if in this case a chemical equilibrium can be established.

→ **Authors:** While we agree that it would be great to further explore the effects of the initial condition (e.g. systematically exploring the effects of TMO enrichment), we feel that it is beyond the scope of this paper, and better suited in a follow-up study. However, as mentioned above, we added few lines in the discussion that describe briefly the influence of B (lines 391-403).

**Reviewer Point 3:** Two effects have been neglected, but they can also result in chemical equilibration and compositional mixing before final magma ocean crystallization: 1) When the solid mantle grows and has no fixed boundaries, as is assumed here, convection causes the new top crystallized layer, which should have a different FeO content, to sink and mix continuously with the solid mantle. 2) If convection in the solid mantle starts before the solidification of MO, partial melting of the cumulates and 'feeding' of the MO with this melt is very likely. Both effects change the chemical equilibrium considerably and do not necessarily require the material to be able to flow through by phase change. However, the latter may further reduce the time scale of chemical equilibrium.

→ **Authors:** These are two important points and we modified the manuscript to address them. Regarding the first one: in this paper we assume fixed boundaries and test different thicknesses of the solid mantle (or magma oceans). We show that chemical equilibration occurs on a timescale that is much smaller than that of crystallisation over a wide range of parameters (lines 421-424). Thus we expect: if we had considered that the solid mantle grows, the newly formed cumulates should have a very similar composition than that modelled here (because of swift equilibration). To explore the effects of the growth of the solid mantle ("moving boundary") systematically will be the subject of future studies (technical development and computationally demanding modelling is required) (lines 329-331).

Regarding point 2: we totally agree. Decompression melting of the convecting cumulates (or compression melting near the BMO-mantle boundary) is exactly the process we have in mind to justify our boundary condition. As we describe in the manuscript, convection supports dynamic topography at the phase change, and this dynamic topography is removed by melting/refreezing according to the phase-change boundary condition. The efficiency of melt-solid segregation near the boundary is one of the processes that is captured by our phase change number $\Phi$. As the effective value of $\Phi$ remains highly uncertain, we explore it over a wide range. We added a few sentences in lines 324-326 to further clarify the implications of the phase-change boundary condition.

**Reviewer Point 4:** Steady-state simulations, i.e. delta T is constant, but also no internal heat sources and a constant viscosity are used – all these effects can influence the strength of convection (and chemical equilibration) and possibly the convection pattern. In particular the influence of internal heat and a temperature dependent viscosity could be tested fast.

→ **Authors:** Indeed these effects can influence the results. But testing the effect of internal heat sources, as well as temperature-dependent viscosity, goes beyond the scope of the current paper. These should be definitely taken into account in future projects. In a pilot study, we are focusing here on the first-order effects of $\Phi$ and Ra (and geometry) on compositional equilibration during magma-ocean crystallisation. Also note that the distribution of heat sources is time-dependent, but our approach here is to run simplified steady-state models.

**Reviewer Point 5:** A new phase change boundary condition for convection has been investigated, are there benchmark studies also for sufficient resolution or how do the authors ensure that the calculations are correct?

$\rightarrow$ **Authors:** The phase change boundary condition implementation was tested against linear stability analysis (Agrusta et al. (2019) for cartesian geometry; Morison (2019) for spherical geometry). In our models, we also check the energy conservation over the solid mantle, and iron mass conservation over the whole mantle (solid mantle and magma oceans). Both checks are successful within machine precision.

**Reviewer Point 6:** A table should be added for all parameters used.

$\rightarrow$ **Authors:** It is now included.

**Reviewer Point 7:** Line 255: It is not clear to me why with small Phi the volume of the solid mantle has no effect on the time scale of chemical equilibration. Do the authors have an explanation? I think this also indicates that the material flow is extremely large - is this really realistic? One could estimate the value.

$\rightarrow$ **Authors:** Indeed, the coefficients of the fitting equation indicate that at low values of $\Phi$, the ratio between volumes (i.e., the aspect ratio of the evolution scenario), has a much smaller impact than at high values of $\Phi$. One explanation for this is that at low values of $\Phi$, convection occurs with low degree, so the geometry of the problem is less important. We clarify this in the new version of the manuscript in lines 268-273. Extremely large material flow is indeed realistic for these low values of $\Phi$, again due to the geometry of flow (at low degrees of convection/translation, near-zero shearing occurs across the solid mantle which is bound by a TMO+BMO) (e.g. Deguen, 2013; Deguen et al, 2013; Morison, 2019; Morison et at, 2019). In terms of which $\Phi$ is realistic for the early Earth: this is uncertain, but see our discussion at lines 140-149.

**Reviewer Point 8:** Line 310: In the discussion, the crustal dichotomies of Mars and the Moon are mentioned and associated with the present process. I don't find this so obvious, because according to the model low degree convection is postulated at the beginning of the MO crystallization, but the crustal dichotomy is more likely to occur at the end of the MO phase, when the pattern becomes small scale.

$\rightarrow$ **Authors:** We modified the text to address this point (lines 334-336).

**Reviewer Point 9:** Line 370: It is stated that smaller planets cool faster. This is not generally true, for example if a blanketing crust is formed during MO crystallization before the mantle is entirely solid and the cooling and crystallization of the MO slows down considerably - as postulated for the Moon.

$\rightarrow$ **Authors:** This is a good point and we added a sentence to address it (lines 405-406).

**Reviewer Point 10:** Line 380: "for realistic values for the phase change number Phi+ smaller than $\sim 100$". I doubt that we really know the realistic value in view of the simplification of the process and the unknown parameters.

$\rightarrow$ **Authors:** We removed the word 'realistic' in this sentence. In terms of the relevance of values of phi, see our discussion in subsection "2.2 Dynamic topography and the phase change boundary condition". We agree that there are large uncertainties.

**Reviewer Point 11:** Figure 7 is difficult to read with the different symbols and lines.

→ **Authors:** We agree that figure 7 may be too complicated due to the large amount of information in it. We remove figure 7 from this manuscript, since the relevant information is summarised in the text, and the good agreement of the scaling law with the predictions is shown in figure A1.

[revised manuscript text omitted]

---

## Author Response (AR3)

**Timescales of chemical equilibrium between the convecting solid mantle and over-/underlying magma oceans**

Daniela Bolrão, Maxim Ballmer, Adrien Morison,
Antoine Rozel, Patrick Sanan, Stéphane Labrosse, Paul Tackley

December 7, 2020

**Response to the Editor**

Once again, we would like to thank the Editor for handling our paper and the Reviewers for taking the time to review our manuscript so thoughtfully a second time. We took the useful comments into consideration and made changes in the manuscript to address them. The reviewers address an important point regarding the value of the Buoyancy number, B. We agree that having B as part of the parameter space would have benefited the paper. We tried to re-run all the cases again but due to technical difficulties, most of simulations didn't reach the end. However, the new preliminary results look similar to the results presented in this paper and we are convinced that the value of B doesn't bring major changes in the timescales of chemical equilibrium between the solid mantle and magma oceans. Unfortunately, these preliminary results have a quantitative error and therefore, cannot be published. We intend to explore the effects of B in our next study.

Once again, we would like to thank the Editor, the Reviewers and the Editorial Board for handling our paper. Please find our responses to the comments of the Reviewers below.
* * *
**Response to Reviewer Number 1**
**General comments**

**Reviewer:** While most points I raised about presentation and explanations have been satisfactorily taken into consideration, my concern on the effect of chemical buoyancy could not be addressed due to technical issues that I can appreciate. This is unfortunate because the article would clearly benefit from including the buoyancy number in its parameter space. However the results on the timescale of equilibration between mantle and magma ocean(s) are still valuable, but the discussion should elaborate more on the the composition of the magma ocean (see specific comments). The paragraph which has been added about the effect of chemical buoyancy is not entirely satisfactory in my opinion, and could point out the limitation of having B=1 rather than trying to convince the reader that the effect of this parameter is not important, relying on (what I found) a shaky argument. Finally, I am also puzzled by the use of the term "chemical equilibrium", which, in its common understanding, is something very different from what is dealt with here. I would suggest "mass" or "composition" equilibrium, which doesn't have as strong an underlying meaning.

**Authors:** Regarding the value of B and discussion elaboration, please see our response to specific comments below. Regarding the term "chemical equilibrium", we don't see a problem in keeping this term.

**Specific comments**

**Reviewer:** *L 110-113: Regarding the buoyancy number, although earth like models point to a value of B∼ 3.0 today, the value in the early Earth might be different. In this paper we make a conservative choice and consider a buoyancy number of B = 1.0, to attribute similar weight to compositional and thermal effects on the density.* Here a short discussion on what B could be in the early Earth (at least whether it might be higher or lower) would be relevant. Besides, I don't really understand in which sense the choice of B=1 is "conservative". A value higher than the expected one (2.8 based a the present-day Earth mantle) would be conservative in the sense that one would be sure not to underestimate the effect of chemical buoyancy. Furthermore, since the XSFeO cannot reach high values (in particular it cannot be higher than XMOFeO, which is lower than 0.25 in the cases presented in Figure 3), the available density contrast in the mantle is reduced, so taking B=1 attributes an effective weight to compositional effects that is significantly lower than the thermal ones (the total temperature range being ensured to be covered due to the boundary conditions).

**Authors:** A discussion about the value of B was already written in the previous version of this manuscript (Discussion section), which explains why B=1 is still a valid choice. We intend to deepen our discussion about the effects of B in our next study.

**Reviewer:** *L 177: We calculate the influx of mass corresponding to this dynamic topography, and distribute this mass by the new tracers* How do you relate the number of tracers to the inflowing mass? Based on their initial density (i.e. number of tracers per cell)? Or do you balance the number of tracers lost at melting boundaries to that of tracers gained at crystallizing boundaries?

**Authors:** We included a more detailed explanation in the new version of the paper.

**Reviewer:** L 208-287: In parts 3.1 and 3.2 you present results in terms of time for half equilibrium for cases having both a 2390-km-thick mantle (either TMO or BMO) and a 2290-km-thick mantle (both BMO and TMO). The time normalization is different for those two cases (one non-dimensional unit time for the former is about 9% longer than one non-dimensional unit time for the later). Since you compare results from both cases and use them to fit the scaling law, they would probably need to be normalized to a common scale; are they? Since the result exhibits variations in orders of magnitude for the tie of half equilibrium, I don't expect the conclusion to change.

**Authors:** Yes, this was taken into consideration and the values were normalised before being used to fit the scaling law.

**Reviewer:** *L 400-402: Thus, the actual value considered for B is expected to have only a minor effect on the equilibration timescales constrained here.* I am not entirely convinced by this reasoning. What you observe is that, for B=1, magma ocean(s) and mantle equilibrate swiftly, diminishing the influence of chemical buoyancy. However, in order to conclude that

the effect of B is minor, you have to assume that equilibration between magma ocean(s) and mantle will be fast regardless of the value of B, which cannot be justified based on the present work.

**Authors:** We expect that with this phase-change boundary condition, large amount of material will cross the boundary (several citations along the paper). We expect that this effect will lead to a swift equilibration between magma ocean(s) and mantle.

**Reviewer:** Here are a couple of ideas to extend the discussion: Your results seem to suggest that the magma ocean crystallizes in the equilibrium crystallization regime (where it is in equilibrium with the whole mantle at all times). In this case, you could easily compute the evolution of XMOFeO and the crystallizing material's XFeO as a function of the magma ocean's volume and be more quantitative about the magma ocean's enrichment. Moreover, you could do the same for lower partition coefficient and extend the discussion to other species (e.g. volatiles: how do your results compare with the assumptions for volatiles partitioning from Lebrun et al., 2013 and the other studies?). I would also suggest, for the available simulations, to study the time evolution of XFeO in the crystallizing material (I guess it can be easily derived from the value of XMOFeO) and to compare it to the average value in the mantle. This could help discussing the effective chemical density contrast.

**Authors:** These are great discussion points that we'll keep in mind for our next study.

**Technical corrections**

**Reviewer:** • *L 271-272: One possible explanation for this is that at low values of* $\Phi$*, convection occurs with low degree, so the geometry of the problem is less important.* Is it a "possible explanation" for the previous statement ("this conclusion is independent of the evolution scenario") or for the weak influence of the solid mantle's volume?

• *L 349: it is conceivable that a thermally-coupled TMO and BMO crystallise more slowly than expected* it is conceivable that thermally-coupled TMO and BMO crystallise more slowly than expected

• *L 394: (i.e., a more realistic initial condition at least for the TMO)* I don't understand what you mean here.

**Authors:** All these comments are very useful and we tried to address them all in the text.
* * *
**Response to Reviewer Number 2**

**Reviewer comment:** The authors have answered most of my questions and comments satisfactorily. I can accept the publication of this study, but I would nevertheless like to emphasize two concerns. For technical reasons they were not able to test the influence of the buoyancy coefficient B, and it is now argued that the value of B=1 used is a conservative choice. I am not entirely convinced for two reasons: Under the assumption of equilibrium crystallisation

as in the current model setup, a large B-value would more strongly impede the downwellings formed at the boundary between TMO and solid mantle, as they are depleted in FeO. On the other hand, if one assumes fractional crystallisation as initial setup, I wonder if the chemical density contrast is not too great to allow efficient convection and mixing when the heavy material is rearranged, i.e. at the bottom of the mantle. However, the authors claim that the preliminary results are similar to the present results, and perhaps that is OK.

**Authors:** We agree that this study would have benefited from the exploration of the value of B. We tried to re-run the simulations with a different values of B and the preliminary results are similar to the ones presented in this paper. We are convinced that this choice of B doesn't change the main conclusions of the paper. We intend to explore the effects of B in our next study to consolidate this matter, and further discuss the influence of B.

**Reviewer comment:** I am also not entirely satisfied with their answer to my comment: "If convection in the solid mantle starts before the solidification of MO, partial melting of the cumulates and 'feeding' of the MO with this melt is very likely. Both effects change the chemical equilibrium considerably and do not necessarily require the material to be able to flow through by phase change. However, the latter may further reduce the time scale of chemical equilibrium." The authors state on line 345' "Without this boundary condition, partial melting of solid cumulates could still change the composition of the magma ocean, but we show that with this boundary condition a larger volume can (re-)melt and crystallise efficiently at either or both solid-liquid phase boundaries. " I wonder how they show or quantify this statement, as they have not considered the melting of the cumulates nor estimated the amount of this melting - or at least it is not shown. I suspect that which effect dominates to obtain chemical equilibrium (i.e. the phase boundary condition or partial melting of the cumulates) depends in turn on the chosen value of the phase change number

**Authors:** We agree that those sentences could be improved. We modified the manuscript to address these comments.

**Reviewer comment:** Please check Eq. 13, I think it should be minus b - at least to get your values on line 225.

**Authors:** We changed the minus in Eq. 13, it was a typo.

[revised manuscript text omitted]
} + \mathrm{Ra}\Big(T - \langle T \rangle - \mathrm{B}(X_{\mathrm{FeO}}^{\mathrm{S}} - \langle X_{\mathrm{FeO}}^{\mathrm{S}} \rangle)\Big)\hat{\mathbf{r}} = 0 \tag{5}$$

with $\mathbf{u}$ the velocity field, $\langle T \rangle$ the lateral average of the temperature field $T$, $t$ the time, $X_{\mathrm{FeO}}^{\mathrm{S}}$ the FeO molar content in the solid mantle, $\langle X_{\mathrm{FeO}}^{\mathrm{S}} \rangle$ the lateral average of $X_{\mathrm{FeO}}^{\mathrm{S}}$, $p$ the dynamic pressure, Ra the Rayleigh number and B the buoyancy number. The last two are defined respectively as:

$$\mathrm{Ra} = \frac{\rho g \alpha \Delta T h_{\mathrm{S}}^3}{\eta \kappa}, \tag{6}$$

$$\mathrm{B} = \frac{\beta}{\alpha \Delta T}. \tag{7}$$

In this study we consider that magma oceans and solid mantle are made only of (Fe, Mg)O (see section 2.3 for more details). We set temperature to 1.0 and 0.0, respectively, at the bottom and top solid domain boundaries. Regarding the buoyancy number, although  Earth-like models point to a value of B $\sim 3.0$ today, the value in the early Earth might be different. In this paper we make a conservative choice and consider a buoyancy number of B $= 1.0$, to attribute similar weight to compositional and thermal effects on the density.

The solid domain is represented using the spherical annulus geometry (Hernlund and Tackley, 2008), composed of a grid of $512 \times 128$ cells, in which Eq. (2) – Eq. (5) are solved. Composition is advected by tracers. We assume that each magma ocean is well-mixed and that its dynamics are fast compared to that of the solid mantle. In our setup, magma oceans are treated as simple 0D compositional reservoirs at solid mantle boundaries. We hereafter use superscripts '$+$' and '$-$' to refer, respectively, to top and bottom solid mantle boundaries. In equations, the sign '$\pm$' reads as '$+$' if a TMO is considered, and '$-$' if a BMO is considered. The subscript '$_{\mathrm{MO}}$' refers to Magma Ocean. Thus, when we introduce a quantity, e.g. $\xi$, related to a magma ocean, we introduce it as $\xi_{\mathrm{MO}}^{\pm}$, with $\xi_{\mathrm{MO}}^{+} = \xi_{\mathrm{TMO}}$ relating to the TMO, and $\xi_{\mathrm{MO}}^{-} = \xi_{\mathrm{
[revised manuscript text omitted]

---

## Author Response (AR4)

**Timescales of chemical equilibrium between the convecting solid mantle and over-/underlying magma oceans**

Daniela Bolrão, Maxim Ballmer, Adrien Morison,
Antoine Rozel, Patrick Sanan, Stéphane Labrosse, Paul Tackley

December 16, 2020
* * *
**Executive Editor Susanne Buiter:**

Dear authors,

Thank you for your revisions and for the attempts to rerun your calculations for a different buoyancy number. I appreciate that several circumstances have finally prevented the inclusion of buoyancy number variations, which is a real pity, but i am happy to follow the advice of the reviewers and topical editor that the models as they are, are substantial enough for publication.

The reason I am asking for a minor revision is to give you the chance to elaborate the discussion on the buoyancy number, as suggested by the reviewers. This should include a short discussion on what B could be in the early Earth (whether it might be higher or lower) and in which sense the choice of B = 1 is conservative (points by reviewer 1). Please also do not postpone a discussion of the first point raised by reviewer 2 to a later manuscript, i think it is warranted to at least discuss what the implications of variations in B could be.

I trust that this addition can be done swiftly and I am looking forward to revised version.

With best wishes, Susanne Buiter

**Authors:**

Dear Susanne,

We would like to thank you for your comments and for giving us the opportunity to make changes in our manuscript. We tried to accommodate all the comments in this new version.

Please see below the changes in the Methods Section and Discussion Section.

All the best,

Daniela Bolrão, on behalf of all co-authors

[revised manuscript text omitted]
} + \mathrm{Ra}\Big(T - \langle T \rangle - \mathrm{B}(X_{\mathrm{FeO}}^{\mathrm{S}} - \langle X_{\mathrm{FeO}}^{\mathrm{S}} \rangle)\Big)\hat{\mathbf{r}} = 0 \tag{5}$$

with $\mathbf{u}$ the velocity field, $\langle T \rangle$ the lateral average of the temperature field $T$, $t$ the time, $X_{\mathrm{FeO}}^{\mathrm{S}}$ the FeO molar content in the solid mantle, $\langle X_{\mathrm{FeO}}^{\mathrm{S}} \rangle$ the lateral average of $X_{\mathrm{FeO}}^{\mathrm{S}}$, $p$ the dynamic pressure, Ra the Rayleigh number and B the buoyancy number. The last two are defined respectively as:

$$\mathrm{Ra} = \frac{\rho g \alpha \Delta T h_{\mathrm{S}}^3}{\eta \kappa}, \tag{6}$$

$$\mathrm{B} = \frac{\beta}{\alpha \Delta T}. \tag{7}$$

In this study we consider that magma oceans and solid mantle are made only of (Fe, Mg)O (see section 2.3 for more details). We set temperature to 1.0 and 0.0, respectively, at the bottom and top solid domain boundaries. Regarding the buoyancy number,  Earth-like models point to a value of  B ≈ 3 for the present-day mantle (i.e., based on the density difference between Mg- and Fe-rich silicate end-members, as well as CMB temperature estimates). However, there are significant uncertainties associated with the value of $\alpha$ (which is temperature and pressure depen and that of $\Delta T$ in the solid part of the primitive mantle. For example, $\Delta T$ increases with the thickness of the solid layer. Within these uncertainties, at least a range of $1 \leq \mathrm{B} \leq 3$ is acceptable. In this paper , we choose B = 1.0  in order to limit the impact of the initial condition on the onset of convection (see discussion in Section 4).

The solid domain is represented using the spherical annulus geometry (Hernlund and Tackley, 2008), composed of a grid of $512 \times 128$ cells, in which Eq. (2) – Eq. (5) are solved. Composition is advected by tracers. We assume that each magma ocean is well-mixed and that its dynamics are fast compared to that of the solid mantle. In our setup, magma oceans are treated as simple 0D compositional reservoirs at solid mantle boundaries. We hereafter use superscripts '+' and '−' to refer, respectively, to top and bottom solid mantle boundaries. In equations, the sign '±' reads as '+' if a TMO is considered, and '−' if a BMO is considered. The subscript 'MO' refers to Magma Ocean. Thus, when we introduce a quantity, e.g. $\xi$, related to a magma ocean, we introduce it as $\xi_{\mathrm{MO}}^{\pm}$, with $\xi_{\mathrm{MO}}^{+} = \xi_{\mathrm{TMO}}$ relating to the TMO, and $\xi_{\mathrm{MO}}^{-} = \xi_{\mathrm{
[revised manuscript text omitted]

---

## Author Response (AR5)

**Timescales of chemical equilibrium between the convecting solid mantle and over-/underlying magma oceans**

Daniela Bolrão, Maxim Ballmer, Adrien Morison,
Antoine Rozel, Patrick Sanan, Stéphane Labrosse, Paul Tackley

December 24, 2020
* * *
**Authors:**
Dear Susanne,

Thank you for accepting our paper! We would like to thank you once again for giving us the opportunity to make changes in the previous version of the manuscript. Once again, we would like to thank the Editor Julien Aubert and the Reviewers for the useful comments that improved the manuscript. We would also like to thank the whole Editorial Board and the Editorial Support Team for handling our paper.

All the best and happy holiday season,

Daniela Bolrão, on behalf of all co-authors